

# Nature-Based Solutions for hydro-meteorological risk reduction: A state-of-the-art review of the research area

Laddaporn Ruangpan[1,2,], Zoran Vojinovic[1,3], Silvana Di Sabatino[4], Laura Sandra Leo[4], Vittoria Capobianco[5], Amy M. P. Oen[5], Michael McClain[1,2] , Elena Lopez-Gunn[6]

[1] IHE Delft Institute for Water Education, Delft, the Netherlands
[2] Department of Water Management, Faculty of Civil Engineering and Geosciences, Delft University of Technology, the Netherlands
[3] College for Engineering, Mathematics and Physical Sciences, University of Exeter, UK
[4] Department of Physics and Astronomy, University of Bologna, Italy
[5] Norwegian Geotechnical Institute, Norway
[6] ICATALIS, Spain

*Correspondence to*: Laddaporn Ruangpan (L.Ruangpan@tudelft.nl)

**Abstract.** Natural hazards such as severe floods, storm surges, landslides, avalanches, hail, windstorms, droughts, heat waves and forest fires occur almost daily. This situation is likely to become worse given the projected changes in climate, degradation

of ecosystems, population growth and urbanisation. The new concepts such as Ecosystem-based Adaptation, Green Infrastructure and/or Nature-Based Solutions have emerged as effective means to respond to such challenges. The present paper provides a critical review of the literature and identifies current knowledge gaps and future research prospects. There has been an explosion of scientific publications on this topic with a more significant rise taking place from 2007 onwards. Hence, the review process started by sourcing 1381 articles from Scopus which were also cross-referenced with the articles

sourced from Web of Science and Google Scholar. The full analysis was performed on 159 closely related articles. The analysis confirmed that numerous advancements have been achieved to date. These solutions have already proven to be valuable in providing sustainable, cost-effective, multi-purpose and flexible means for hydro-meteorological risk reduction. However, there are still many areas where further research and demonstration is needed in order to promote their upscaling and replication and to make them become mainstream solutions.

# 1 Introduction

There is an increasing evidence that climate change and associated hydro-meteorological disasters are already causing wide-ranging impacts on different sectors of society. Natural hazards such as severe floods, storm surges, landslides, avalanches, hail, windstorms, droughts, heat waves and forest fires have already made unprecedented impacts on the global economy, human well-being, and the environment. Some of the main reasons for this situation are climate change, land use change, water use change

and other pressures linked to population growth (Thorslund et al., 2017a) and the situation is likely to become worse given the projected changes in climate (see for example, EEA, 2017). Therefore, effective adaptation strategies are needed to mitigate risks



related to the increased frequency of extreme events (Maragno et al., 2018). Since biodiversity and ecosystem services can play important role in responding to climate-related challenges, both mitigation and adaptation strategies should take into consideration a variety of Green Infrastructure (GI) and Ecosystem-based Adaptation (EbA) as effective means to respond to present and future disaster risk (see also EEA, 2015). Such approaches are already well accepted within multilateral frameworks such as the

international United Nations (UN) Framework Convention on Climate Change (UNFCCC), the Convention on Biological Diversity (CBD) and the Sendai Framework for Disaster Risk reduction (SFDRR). As such, they are recognized as effective means for climate change adaptation (CCA) and disaster risk reduction (DRR), and for the implementation of the Sustainable Development Goals (SDGs).

In view of the above, many countries are nowadays developing adaptation and mitigation strategies based on GI and EbA to

reduce their vulnerability to hydro-meteorological hazards (Rangarajan et al., 2015). Nature-Based Solutions (NBS) have been introduced relatively recently.  The reason behind is that NBS offer the possibility to work closely with nature to adapt with the future changes and to reduce the impact of climate change as well as to improve human well-being (Cohen-Shacham et al., 2016).  NBS have been in the focus for reseach in several EU Horizon2020 funded projects. The Horizon2020 offers new opportunities in the focus area of 'Smart and Sustainable Cities with Nature based solutions' (Faivre et al., 2017). Some of

these important projects are: Nature4Cites, Naturvation, NAIAD, BiodiverEsA, Inspiration, URBAN GreenUP, UNaLaB, URBINAT, CLEVER Cities, proGIreg, EdiCINET, RECONECT, OPERANDUM, ThinkNature,  EKLIPSE and PHUSICOS (nature4cities, 2019).

NBS are typically implemented through both structural (green-blue infrastructure, e.g. wetlands, green roofs) and non-structural measures (e.g. improving the local knowledge of NBS) (Lottering et al., 2015; Raymond et al., 2017). They are

associated with multiple benefits such as improving water quality, increasing the opportunities for recreation, improving human well-being and health, enhancing vegetation growth and connecting habitat and biodiversity (Donnell et al., 2018; Raymond et al., 2017; Song et al., 2018; Thorslund et al., 2017b).

The number of scientific studies focused on GI, EbA and/or NBS to reduce disaster risk are continuously increasing all over the world. The aim of this article is to provide a state-of-the-art review of publications on hydro-meteorological risk reduction

with NBS and to explore the patterns and trends of current research activities as well as to indicate some directions for future research based on the knowledge gaps. The systematic review process presented in this article concerns only scientific journal articles although there is a considerable body of knowledge available in various project reports and other kind of literature. However, since they do not necessarily follow scientific publication standards most of them were excluded from the scope of the present work. Only in those cases where with a more significant contribution has been achieved (and in the absence of

scientific articles) such literature was included into the analysis. The key objectives of the present review work are as follows:



1) To identify patterns and trends of NBS publications in scientific journals,
2) To assess the state-of-the-art in research concerning both small and large scale NBS,
3) To review the use of techniques, methods and tools for planning, selecting, evaluating and implementing NBS,
4) To review the socio-economic influence in the implementation of NBS as well as their main benefits and co-benefits, and
5) To identify knowledge gaps and proposed future research prospects.

## 2 Materials and methodology

### 2.1 Search strategy

The review analysis concerned articles from scientific journals written in English. Two main concepts were used in the search: Nature-Based Solutions and hydro-meteorological risk. As the concept of 'Nature-Base Solutions' appears under different names (which more or less relate to the same field of research), articles related to Low Impact Developments (LIDs), Best Management Practices (BMPs), Water Sensitive Urban Design (WSUD), Sustainable Urban Drainage Systems (SuDS), Green Infrastructure (GI), Blue-Green Infrastructure (BGI), Ecosystem-based Adaptation (EbA) and Ecosystem-based Disaster Risk Reduction (Eco-DRR) were included in the identification of relevant articles (see Table 1). The review of hydro-meteorological risk included literature on relevant terms (i.e. disaster, review, hydrology etc.) and different types of risk (i.e. floods, droughts, storm surges and landslides, and the relevant terms.) (Table 1).

During the construction of the queries, the strings were searched only within Index terms and Metadata "titles, abstract, and keywords" in the Scopus database. The search terms for the two concepts were linked with the Boolean operator "AND" while the Boolean operator "OR" was used to link between the possible terms (Table1). An example of a protocol is shown below:

"TITLE-ABS-KEY ( "Nature-based solutions" OR "Nature based solutions" OR "Nature Based Solutions" OR "Nature-Based Solutions" OR "Low impact development" OR "Sustainable Urban Drainage Systems" OR "Water Sensitive Urban Design" OR "Best Management Practices" OR "Green infrastructure" OR "Green blue infrastructure" AND "flood" ) AND ( LIMIT-TO ( DOCTYPE , "ar" ) OR LIMIT-TO ( DOCTYPE , "ch" ) OR LIMIT-TO ( DOCTYPE , "re" ) OR LIMIT-TO ( DOCTYPE , "bk" ) ) AND ( LIMIT-TO ( LANGUAGE , "English" ) )"

Figure 1. shows the number of articles that have been published on the concepts of NBS, LIDs, SuDS, WSUD, BMPS, GI, and BGI. From Fig. 1, it can be observed that since 2007 the number of scientific articles started increasing significantly. Therefore, the time window selected for the review process was from 2007 onwards.

The findings from Scopus were cross-referenced with other databases such as Web of Science and Google Scholar and the number of publications found in Scopus database was larger than the number of articles found in the other two databases.



## 2.2 Selection process

As discussed in Section 2.1, the search process was based upon the following three criteria: (1) articles published in peer-reviewed, scientific journals written in English; (2) articles reported on NBS in terms of hydro-meteorological risk reduction; (3) articles published from year 2007 onwards.

Initially, the Scopus database search resulted in 1381 articles published in scientific journals. The same search performed in Web of Science and Google Scholar resulted in 1208 and 972 articles, respectively. Hence, the Scopus database was used as a main database for the purposes of the present work. To make the review process more specific, the process depicted in Fig. 2 was applied to select the relevant articles. Firstly, those duplicate articles found from the applied queries were removed. After that, all articles were analysed on the basis of their titles and keywords. Since the search of articles contained some gaps (i.e.,

there were several missing articles which were already known to the authors) the list of articles was appended with those missing articles. The final step was to read and analyse all selected articles.

## 3 Overview of definitions and theoretical backgrounds

There are several terms and concepts which have been used interchangeably in the literature to date. In terms of NBS, the two most prominent definitions are from International Union for Conservation of Nature (IUCN) and the European Commission.

The European Commission defines Nature-Based Solutions as "*Solutions that aim to help societies address a variety of environmental, social and economic challenges in sustainable ways. They are actions inspired by, supported by or copied from nature; both using and enhancing existing solutions to challenges, as well as exploring more novel solutions. Nature-based solutions use the features and complex system processes of nature, such as its ability to store carbon and regulate water flows, in order to achieve desired outcomes, such as reduced disaster risk and an environment that improves human well-being and*

*socially inclusive green growth*" (European Commission, 2015). The IUCN has proposed a definition of NBS as "*actions to protect, sustainably manage and restore natural and modified ecosystems that address societal challenges effectively and adaptively, simultaneously providing human well-being and biodiversity benefits*" (Cohen-Shacham et al., 2016).

Nature-Based Solutions comprise a wide range of perspectives and frameworks with differences in the way they are specified. In the same manner that the involvement of NBS can be diverse, as they have been called differently in the way their

participation is found. In this regard, as indicated earlier seven different terminologies were found in the scientific literature. The timeline of each terminology based on their appearance in scientific journals is shown in Figure 3 and their definitions are given in Table 2.

The common idea behind these terms is the use of landscape for transforming the linear character of conventional stormwater management into a more cyclic approach where drainage, water supply, and ecosystems are treated as part of the same system,

mimicking more natural water flows (Liu and Jensen, 2018). The literature to date defines these solutions are more or less




equally effective in addressing climate change adaptation and disaster risk reduction. The synergy between GI, BGI, EbA, Eco-DRR, and NBS is that they take a participatory, holistic, integrated approach using nature to enhance adaptive capacity, reduce disaster risk, reduce the vulnerability, increase the resilience, enhance biodiversity, and improve human well-being. More information on the history, scope, application and underlying principle of terms of SuDs, LIDs, BMPs, WSUD and GI

can be found in Fletcher et al., (2015).

Although they are all based on a common idea, differences in definition among these terms can be explained from their historical perspectives and knowledge base pertinent for that point in time (Fletcher et al., 2015). The distinguishing characteristic of EbA, Eco-DRR, and NBS is how they addresses social, economic and environmental challenges (Faivre et al., 2018). EbA focuses more on a long-term change within the conservation of biodiversity, ecosystem services, and climate

change, while Eco-DRR is more focused on immediate and medium-term impacts from the risk of weather, climate and no climate-related hazards. EbA is often seen as a subset of NBS that is explicitly concerned with climate change adaptation through the use of nature (Kabisch et al., 2016). Different from these two terms, NBS offer an integrated way to look at different issues simultaneously. However, it is important to note that very often a combination between natural and traditional engineering solutions (a.k.a. "hybrid" solutions) is likely to produce more effective results than any of these measures alone,

especially when their co-benefits are taken into consideration (Alves et al., 2019).

Important advance in the science and practice of NBS is given by the EKLIPSE Expert Working Group, who developed the first version of multiple-dimension impact evaluation framework to support planning and evaluation of NBS projects. The document includes a list of impacts, indicators and methods for assessing the performance of NBS in dealing with some major societal challenges (EKLIPSE, 2017). The framework is based on 10 challenges: 1) Climate Mitigation and adaptation, 2)

Water Management, 3) Coastal Resilience, 4) Green space Management, 5) Air Quality, 6) Urban Regeneration, 7) Participatory Planning and Governance, 8) Social justice and Social Cohesion, 9) Public health and well-being and 10) Economic opportunities and Green Jobs (Raymond et al., 2017). The fact that the EKPLISE framework was specifically develop for NBS at the urban scale and only deals with challenges faced by cities. Lafortezza et al., (2018) reviewed different case studies around the world where NBS have been applied from micro-scale to macro-scale.

In the recently commissioned European Community funded RECONECT (Regenerating ECOsystems with Nature-based solutions for hydro-meteorological risk rEduCTion) project, one of the main objectives is to address performance of NBS at different scales, contexts and hybrid combinations with grey infrastructure (RECONECT, 2019). RECONECT project connects to some of the work and case studies undertaken in the FP7 PEARL project (see PEARL, 2019a; Vojinovic, 2015)



## 4 Findings

### 4.1 Trends, knowledge gaps and future research prospects

The literature material reviewed in this study showed that NBS have not been equally applied to all hydro-meteorological risk reduction contexts. The review identified in total 1381 Journal articles from 2007 to the end of 2018. The patterns of all terminologies of NBS were analysed using 166 publications for hydro-meteorological risk reduction. An overview of some research gaps and future research prospects is given in Table 3.

Most of the literature to date is about NBS in urban areas whereas those contexts concerning river and coastal floods, droughts and landslides are the least addressed. 88% of all articles were concerned with runoff reduction or flood risk reduction in urban areas (Fig. 4a.). Also, only 62 out of 1381 articles (i.e., 4.5%) explicitly used the term "Nature-Based Solution" for hydro-meteorological risk reduction. This can be explained due to difference in terms used in different countries while the term NBS has been used only from 2008 (Fig. 3). However, the significant increase of published articles in recent years testifies how NBS is a rapidly growing research area (Fig. 4b).

In terms of the other relevant literature (i.e., literature that is not published in scientific journals but found to be relevant for the subject matter) the following documents were identified: EKLIPSE, 2017; Asian Development Bank, 2016; Sekulova and Anguelovski, 2017; Kabisch et al., 2017; Renaud et al., 2016.

### 4.2 Small and large scale NBS for hydro-meteorological risk reduction

In this review, NBS for hydro-meteorological risk reduction have been divided into small and large scale (Fig.5). "Small scale NBS" are usually referred to as NBS at the urban or local scale, while NBS in rural areas, river basins and at the regional scale are referred to as "Large scale NBS" (Fig.5.).

### 4.2.1 Small scale NBS

Small scale NBS are usually applied to a specific location such as a single building or a street. However, many researchers argue that a single NBS is not sufficient to control large amount of runoff. Therefore, this review discusses on application and effectiveness of both individual NBS and multiple-NBS combinations.

### (1) Effectiveness of a single/individual NBS site

To date, various types of single NBS sites have been studied with objectives such as reduction of the flood peak (Carpenter and Kaluvakolanu, 2011; Ercolani et al., 2018; Liao et al., 2015; Mei et al., 2018; Yang et al., 2018), delay/attenuation of the flood peak (Ishimatsu et al., 2017), reduction of volume of combined sewer overflows (Burszta-Adamiak and Mrowiec, 2013) and reduction of the surface runoff volume (Lee et al., 2013; Shafique and Kim, 2018).



Shafique et al., (2018) described how porous pavement can be very effective in decreasing the possibility of flash floods in the developed area in Seoul. NBS has also been used to improve water quality in Greater Melbourne, Australia (Khastagir and Jayasuriya, 2010) and in Xi' city in China (Li et al., 2018b).

The most common NBS measures in urban areas appear to be intensive green roofs (Burszta-Adamiak and Mrowiec, 2013; Carpenter and Kaluvakolanu, 2011; Ercolani et al., 2018), extensive green roofs (Cipolla et al., 2016; Lee et al., 2013), rain gardens (Ishimatsu et al., 2017), rainwater harvesting (Khastagir and Jayasuriya, 2010), dry detention ponds (Liew et al., 2012), permeable pavements (Shafique et al., 2018), bio-retention (Khan et al., 2013; Olszewski and Allen, 2013), vegetated swales (Woznicki et al., 2018) and trees (Mills et al., 2016). However, the authors of these studies investigated the performance of such measures individually (i.e. at the specific/local/single site) without evaluating it in combination with other NBS sites or in hybrid combinations.

NBS may benefit people in coastal areas by reducing risk from storm surges, wave energy, coastal flooding as well as erosion, as documented by several authors (see for example, Coppenolle, 2018; Joyce et al., 2017; Ruckelshaus et al., 2016; Sutton-Grier et al., 2018). NBS for coastal areas can be implemented either large or small scale. They include dunes, beaches, oyster and coral reefs, mangroves, seagrass beds, and marshes. These measures can also provide habitat for different species such as fish, birds, and other wildlife (Ruckelshaus et al., 2016). However, only few articles focused on the potential benefits of NBS in coastal areas.

The review found just one article that discusses the use of NBS for reduction of drought risk. Lottering et al., (2015) discussed the effectiveness of NBS on reduction of water consumption in suburb areas.

The literature to date acknowledges that the effectiveness of NBS greatly depends on the magnitude and frequency of rainfall events. Green roofs are recognized in reducing peak flows more effectively for smaller magnitude frequent storms than for larger magnitude infrequent storms (see for example, Ercolani et al., 2018). There are also reports that rain gardens are more effective in dealing with small discharges of rainwater (Ishimatsu et al., 2017). Swales and permeable pavements are more effective for flood reduction during heavier and shorter rainfall events. As noted by Qin et al., (2013), small practices may be not sufficient for long duration storm events with consistent rainfall. Hence, a large scale NBS could be a solution for storm events with large magnitude and long duration, which is usually the case for disaster risk reduction applications, and therefore the research in this direction is highly desirable.

Many studies recommend that there is a need to connect an individual NBS with other NBS measures (i.e., a train of NBS) to achieve better runoff control and treat more pollution (see for example, De Risi et al., 2018a; Shafique et al., 2018), to enable more effective long term strategies and to provide a more robust response to larger events with multiple benefits (Webber et al., 2018). Also, Zölch et al., (2017) suggested that the effectiveness of NBS should be directly linked to its ability of increasing





as much as possible the storage capacities within the area of interest, while using open spaces that have not been used previously and/or while providing benefits to other areas for urban planning

**(2) Effectiveness of multiple NBS sites**

There are several studies which have evaluated the performance of multiple (or combined) NBS measures (see for example, Damodaram et al., 2010; Earthman et al., 2012; Huang et al., 2014; Luan et al., 2017). One of the most successful international projects in combining several NBS measures at the urban scale is the "Sponge City Programme (SCP)" in China. The SCP project was commissioned in 2014 with the aim to implement both concepts and practices of LIDs/NBS as well as various comprehensive urban water management strategies (Chan et al., 2018). Nowadays, SCP is wildly used as the concept ('Sponge City') for a city that needs to increase resilience to climate change. It also includes combination of several systems such as source control system, urban drainage system, and emergency discharge system.

Porous pavement appears as one of the most popular measures suitable to combine with other NBS for urban run-off management. Examples of this are described in Behroozi et al., (2018) who selected swales and porous pavement to reduce peak flow and mean Total Suspended Solids (TSS) concentration. Hu et al., (2017) used inundation modelling to evaluate the effectiveness of rainwater harvesting and pervious pavement as retrofitting technologies for flood inundation mitigation at urbanized watershed. Damodaram et al., (2010) concluded that rainwater harvesting and permeable pavement is likely to be more effective than pond storage for small storms, while the pond is likely to be more effective to manage runoff from the more intensive storm.

Several studies argue that multiple NBS measures can lead to a more significant change in runoff regime than single NBS measures For example, eight scenarios were simulated by changing the percentage of combined green roof and permeable pavement in an urban setting (Wu et al., 2018) . The results show that for a scenario where green roof and permeable pavement were applied at all possible locations, a 28% reduction in maximum inundation can be obtained. In comparison, scenarios implementing either green roof or permeable pavement alone at all possible areas experienced a reduction of 14%. One of the main reasons the superior performances of combined NBS is that they are able to work in parallel, each treating a different portion of run-off generated from the sub-catchment (Pappalardo et al., 2017). For these combinations, the spatial distribution should be carefully considered because it can improve the runoff regime better when compared to centralised NBS (Loperfido et al., 2014).

Further research on the use of combined green and grey infrastructures (i.e., hybrid measures) is highly desiderative. To date only three contributions were found in the review. Alves et al., (2016) presented a novel method to select, evaluate and place different hybrid measures for retrofitting urban drainage systems. However, only fundamental aspects were touched in the methodology and they suggested that future work should include possibility of considering stakeholders' preferences or flexibility within the method. Onuma and Tsuge, (2018) compared the cost-benefits and performance between NBS and grey





infrastructures, concluding that NBS are likely to be more effective when implemented through cooperation with local people, whereas hybrid solutions are more effective than a single NBS in terms of performance.

The first limitation of the above studies is that they only assess the effectiveness at urban scales. This may not be sufficient for large events as climate change is likely to increase the frequency and intensity of future events. Large scale NBS may provide

a more significant impact in different management scenarios than just for an urban watershed (Giacomoni et al., 2012). Although Fu et al., (2018) analysed variations in runoff for different scales and land-uses, the impact of NBS was only examined for the small urban scale. There is only one article that deals with hybrid measures (i.e., NBS/green infrastructure and grey infrastructure) and also with combinations of small and large scale NBS. In the work of Vojinovic et al., (2017), a methodological framework that combines ecosystem services (flood protection, education, art/culture, recreation and tourism)

with economic analysis for selection of multifunctional measures and consideration of small and large scale NBS has been discussed for the case of Ayutthaya in Thailand. The third limitation is that none of these contributions have incorporated cost-benefit analyses (CBA). CBA can be used as a tool to support the decision-making process as they serve the feasibility of implementing cost and the potential benefits of NBS.

### 4.2.2 Large-scale NBS

Large-scale water balance, water fluxes, water management and ecosystem services are affected by future changes such as climate change, large-scale land use changes, water use changes and population growth. Therefore, to address such challenges, large scale NBS are needed to make more space for water to retain, decelerate, infiltrate, bypass, and discharge (Cheng et al., 2017; Thorslund et al., 2017b). Generally, a large-scale NBS combines different NBSs within a larger system to achieve better long-term strategies. There are some examples of NBS measures for DRR which are summarized in McVittie et al., (2018).

There are very few articles that have addressed combined behaviour of NBS at the large or catchment scale (see also Table 3). One of the possible reasons is that large-scale systems are much more complex than small-scale systems. The most common large-scale NBS are wetlands (Thorslund et al., 2017b), river restoration (Chou, 2016), flood storage basins (De Risi et al., 2018b), preservation and regeneration of forests in flood-prone areas (Bhattacharjee and Behera, 2018) and making more room for the river (Asselman and Klijn, 2016; Klijn et al., 2018).

A classic example of a large-scale NBS implementation is the 'Room for the River Programme' which was implemented along the Rhine and Meuse rivers in the Netherlands (Klijn et al., 2018). The Room for the River Programme consisted of 39 local projects based on nine different types of measures (Klijn et al., 2013). These measures are flood plain lowering, dike relocation, groyne lowering, summer bed deepening, water storage, bypass/floodway, high water channels, obstacles removing and dikes strengthening. The benefits that the programme achieved are more than just reducing the flooding but also increasing

opportunities for recreation, habitat and biodiversity in the area (Klijn et al., 2013).





Another case study of a large scale NBS is the Laojie river project in Taoyuan City in Taiwan. The study focused on changing the channelised culverted flood-control watercourse into an accessible green infrastructure corridor for the public (Chou, 2016). The landscape changes resulting from this project have increased recreation activities and improved the aesthetic value in the area.

To reduce the impact of large-scale hydro-meteorological events, more research is needed on large-scale NBS and their hybrid combinations designed to attenuate flows and improve drainage. They should be implemented to include improvements in solid waste management, community-based river cleaning programs and reforestation (De Risi et al., 2018b).

To fill this gap, in addition to RECONECT other two "sister" Horizon 2020 projects namely PHUSICOS and OPERANDUM were initiated in 2018 to fill the gap in innovation of NBS and to test their efficacy in rural, mountain and in transition lands

environments. Specifically, PHUSICOS's main aim is to implement and evaluate NBS at regional scale in three large-scale demonstration sites representative of the typical hazards (floods, droughts, landslides) throughout rural and mountainous regions in Europe. OPERANDUM's main aim is to demonstrate viability of NBS in ten sites in Europe, China and Australia including the testing at coastal areas were coastal erosion and storm surge may occur in present and future climate scenarios. Development of techniques, methods and tools for planning, selecting, evaluating and implementing NBS are among the

common products of RECONECT, PHUSICOS and OPERANDUM

### 4.3 Techniques, methods and tools for planning, selecting, evaluating and implementing NBS

Fig. 6. illustrates a typical process for selection and evaluation of NBS (see, for example, (Alves et al., 2016a, 2018). The process starts by selecting possible measures that correspond to the local characteristics and project's target. The next step is concerned with evaluating their performance by numerical models, cost-benefit analysis and/or multi-criteria analysis.

However, for more complex system such a large number of scenarios and parameters, optimisation can be used to maximise the benefits and minimise the costs. The processes above are possible to combine in one tool or to use combination of existing tools to select and evaluate NBS. The techniques, methods and tools for planning, selecting, evaluating and implementing NBS that have been used are reviewed in the following section.

### 4.3.1 Selection of NBS based on local constraints

To date, it has been a well-accepted fact that not all NBS are suitable for all conditions. Therefore, it is important to consider the feasibility and constraints at the site at an early stage in the selection process. The first consideration in selecting NBS is to define the objective such as the target area (i.e. urban, rural) and performance requirements such as quantity and/or quality (Romnée and De Herde, 2015; Zhang and Chui, 2018). For example, Pappalardo et al., (2017) chose permeable pavements and green roofs because they can detain runoff or infiltrate it to the subsoil. Many authors suggest restricting the choice of

appropriate NBS based on common site constraints such as land use, site characteristics (i.e. soil type, groundwater depth,



depth to bedrock), catchment characteristics, political and financial regulations, amenities, environmental requirements and space available (Chen et al., 2013; Eaton, 2018; Joyce et al., 2017; Nordman et al., 2018; Oraei Zare et al., 2012). For example, Eaton (2018) selected bio-retention measures because these are more suitable in low-density residential land use.

Therefore, a screening analysis is necessary to select the NBS measures that are best suited to local constraints and objectives,
providing decision-makers with valuable information  Also, the study of Reynaud et al., (2017) describes how the type of NBS has an impact on individuals' preference for ecosystem services.

The way forward in the selection of NBS is to consider spatial planning principles to locate the position for measures. Spatial planning principles can facilitate and stimulate discussion among local communities, researchers, policy-makers and government authorities.

**4.3.2 Frameworks and methods for evaluation of NBS**

There are several frameworks and methods that can be used to evaluate performance indicators of NBS that are discussed in this review.  One of the most popular evaluation approach is to analyse, simulate and model hydrology, hydraulics and water balance processes. This information is then used to support decision makers, planners and stakeholders in their evaluation performance and potential of NBS by comparing modelled results against current situation, baseline scenario or targets (Jia et
al., 2015). The Curve Number infiltration method can also be used to estimate rainfall runoff based on ground coverage, soil type and precipitation (Maragno et al., 2018).

In addition to the hydrological and hydraulic analysis, cost-benefit analysis is often used to select and implement a cost-effective NBS (Huang et al., 2018; Nordman et al., 2018; Watson et al., 2016; Webber et al., 2018). The common benefits considered include prevented damage costs, omitting infrastructures, profit loss to businesses, prevented erosion damage, and
prevented agricultural losses. One cost-benefit approach is to evaluate NBS by applying the whole life cycle costing approach (LCC) including construction, operation, maintenance and opportunity costs (Nordman et al., 2018) and Return on Investment (ROI) (De Risi et al., 2018b).

An alternative method for evaluation of NBS is multi-criteria analysis (MCA), which has the potential to integrate and overcome the differences between social and technical approaches, (Loc et al., 2017).  It allows to structure complex issues
and help a better comprehension of costs and benefits. Such analysis is useful for decision makers when there are multiple and conflicting criteria to be considered (Alves et al., 2018; Loos and Rogers, 2016). The MCA takes different criteria into account and assigns weights to each criterion. This process can produce ranking of the different measures that can be implemented on the site (Chow et al., 2014; Jia et al., 2015). For examples, Loc et al., (2017) who integrated the results from numerical modelling and social survey into the MCA and ranked the alternatives based on evaluation criteria, which are flood mitigation,
pollutant removal and aesthetics. Loos and Rogers, (2016) applied multi-attribute utility theory (MAUT) to assess utility values for each alternative by assuming that preference and utility are independent from each other. Petit-Boix et al., (2017)



recommended that future research should combine the economic value of the predicted material and ecological damage, risk assessment models and environmental impacts of NBS.

Since not all assessments can be done with modelling alone, interviews and fieldwork are often neccessary. For instance, Chou (2016) used eighteen open questions from six topics, namely: accessibility; activities; public facilities; environmental quality;

ecological value; and flood prevention. These questions are used to evaluate the qualitative performance of river restoration. However, some of the methods are only appropriate for small scale applications and cannot be applied in large catchments. Yang et al., (2018) proposed Relative Performance Evaluation (RPE) methods, which use a score to calculate the performance for all alternatives. This score is calculated as the weighted sum of the scores of individual indicators.

From the discussion above, it can be observed that there are still challenges in evaluating intangible benefits of NBS and

incorporating stakeholders' preferences into the process. For complex systems with a large number of scenarios and parameters, simple trial-and-error methods may not be the feasible approach. In such cases, an automated optimisation method could be effectively applied to handle these tasks and to combine the above mentioned methods. There is also a challenge in combining a range of aspects that can and cannot be expressed in monetary terms into the same framework of analysis.

### 4.3.3 Optimal configuration of NBS

In order to implement NBS, typical selection factors include the number of NBS measures, size, location, and potential combinations of NBS. Optimisation of NBS strategies has been increasingly used in the urban stormwater management context. Most of the studies to date focus on minimising water quantity and improving water quality by selecting the type, design, size and location of NBS (Behroozi et al., 2018; Gao et al., 2015; Giacomoni and Joseph, 2017; Zhang and Chui, 2018). Zhang and Chui (2018) have systematically reviewed optimisation models that have different structures, objectives and

allocation components. This section reviews some examples of using optimisation to assess NBS.

### (1) Comprehensive modelling systems

A comprehensive modelling system typically refers to an optimisation package tool that integrates an "easy-to-use" user interface with physically based deterministic models. Examples include SUSTAIN (the System for Urban Stormwater Treatment and Analysis IntegratioN) (Zhang and Chui, 2018) and Best Management Practice Decision Support (BMPDSS)

(Gao et al., 2015). The SUSTAIN model was developed by the United States Environmental Protection Agency (US EPA) and it aims to provide decision makers with support in the process of selection and placement of NBS measures, and to optimise the hydrological performance and cost-effectiveness of NBS in the urban watershed (Leslie et al., 2009; Li et al., 2018a). There are several studies that apply SUSTAIN with the attempt to minimise the cost of NBS for both runoff quantity (flow volume, peak flow) and runoff quality (pollutant removal) (Gao et al., 2015; Li et al., 2018c).





It is however important to note that comprehensive modelling systems are not always easy to modify to fit with specific needs of users.

**(2) Tools based on integration between optimization algorithms and numerical models**

Integrated model-algorithm tools combine numerical (hydrological-hydrodynamic) models with optimisation algorithms. A popular optimisation method used to evaluate NBS performance is a multialgorithm, genetically adaptive multiobjective (AMALGM) method using the multilevel spatial optimisation (MLSOP) framework (Liu et al., 2016). AMALGM includes Non-dominated Sorting Genetic Algorithm II (NSGA-II), Adaptive metropolis search (AMS), particle swarm optimisation (PSO), and differential evolution (DE) (Wang et al., 2015).

In the reviewed articles, NSGA-II is used in most of the studies to date. Wang et al., (2015) concluded that NSGA-II remains as one of the most popular multiobjective evolutionary algorithms (MOEAs) even with limited parameter tuning, and generally outperformed the other MOEAs concerning the number of solutions contributing to the best-known nondominated set of each problem. There are several examples of the use of NSGA-II. Oraei Zare et al., (2012) minimised run-off quantity while maximizing the improvement of water quality and maximising reliability. Karamouz and Nazif, (2013) minimised cost of flood damage as well as minimising BMP cost in order to improve system performance in dealing with the emerging future condition under climate change impact, Yazdi and Salehi Neyshabouri, (2014) optimised cost-effect, which focused on land use change strategies including orchard, brush and seeding measure in a different part of the watershed. All of the above mentioned studies coupled NSGA-II with the Storm Water Management Model (SWIMM) developed by US EPA (Cipolla et al., 2016; Li et al., 2018b; Mei et al., 2018; Tao et al., 2017; Wu et al., 2018; Yang et al., 2018; Zhu and Chen, 2017) to address the optimisation problems.

There are two different optimisation methods of Particle Swarm Optimization (PSO) which have been found in the course of this review. The modified Particle Swarm Optimization (NPSO) is used by (Duan et al., 2016) to solve the Multi-Objective Optimal (MOO) of the cost-effectiveness of NBS based detention tank design. Similarly, Behroozi et al., (2018) used the multi-objective particle swarm optimisation (MOPSO) to deal with multi-objective optimisation problem by coupling it with SWMM to optimise the peak flow and mean TSS concentration reduction by changing the combinations of NBS.

Another algorithm that is used for optimising the performance of NBS is Simulated Annealing (SA) (Kirkpatrick et al., 1983). SA is a general probability optimisation algorithm that applies thermodynamic theories in statistics. An example of a study with SA is given by Huang et al., (2018) who automatically linked SA with SWMM to maximise cost-benefit for flood mitigation and layout design. The cost-benefit analysis is computed using annual cost, which includes both annual fixed cost and annual maintenance cost. Another study that applied SA is Chen et al., (2017) who combined SA with SWMM to locate



NBS in Hsinchu County in northern Taiwan by considering three objective functions. These were minimising depths, durations, and the number of inundation points in the watershed.

It can be observed that most of the optimisation models to date (both comprehensive modelling system and model algorithms) are coupled with SWMM for urban storm management. There is still a lack of research that uses optimisation to maximise the

efficiency of NBS on a large scale as well as combining other co-benefits in optimisation (Table 3). Furthermore, there is a lack of research that employs 2 dimensional models in the optimisation analysis. This is particularly important when considering estimation of flood damages and other flood propagation-related impacts.

### 4.3.4 Tools for selection, evaluation and operation of NBS

Recently, several selection and evaluation tools (both standalone and web-based) have been developed in order to assist

stakeholders in screening, selecting and visualising NBS measures. Examples of web-based applications which are developed to screen urban NBS measures are Green-blue design tool (atelier GROENBLAUW, 2019), PEARL KB (Karavokiros et al., 2016; PEARL, 2019b), Climate Adaptation App (Bosch Slabbers et al., 2019) and Naturally resilient communities solutions (Naturally Resilient Communities, 2019). These web-based tools allow the user to filter NBS in relation to their problem type, measure, land use, scale, and location.

In addition to the above, there are also tools that combine both the selection and evaluation processes together. An example of such tools which is used to evaluate the performance of NBS is SuDS selection and location (SUDSLOC) tool, which is a GIS tool linked to an integrated 1D hydraulic sewer model and a 2D surface model. Planning-support tool is known as UrbanBEATS (the Urban Biophysical Environments and Technologies Simulator), which aims to support the planning and implementation of WSUD infrastructure in urban environments (Bach et al., 2018). Other tools that can be used to select and

evaluate potential NBS interventions are Long-Term Hydrologic Impact Assessment-Low Impact Development (L-THIA-LID) (Purdue University, 2019) in a web-based application  (Ahiablame et al., 2012; Liu et al., 2015)  and the GIS-based tool called Adaptation Support Tool (AST) (van de Ven et al., 2016; Voskamp and Van de Ven, 2015). Although these tools could be useful in assisting decision makers, some of them may not be suitable for every location and scale. For example, source data required into L-THIA-LID cover only United States and QUADEAU (Romnée and De Herde, 2015) is only suitable for

urban stormwater management in a public space scale.

In addition to the above, other models such as Model of Urban Sewers (MOUSE), nowadays known as MIKE URBAN, MIKE FLOOD and MIKE SHE developed by DHI (Semadeni-Davies et al., 2008), Soil and Water Assessment (SWAT) (Cheng et al., 2017), IHMORS (Herrera et al., 2017), and Urban Water Optioneering Tool (UWOT) (Rozos et al., 2013) can be effectively used in the analysis of NBS.

To date, only few tools have been developed to calculate multiple benefits of NBS in monetary terms as well as to address their qualitative benefits.  Some examples are Benefits of SuDS Tool (BeST), which  is an Excel-Based decision-support tool





that provides a structured approach to evaluating potential benefits of NBS (Digman et al., 2016; Donnell et al., 2018), Blue-Green Cities toolbox which is a GIS toolbox to evaluate multi-benefits, including flood damage reduction, water quality, attractiveness, property prices, habitat size, carbon dioxide sequestration, and reduction in air and noise pollution (BGC, 2016), and the MUSIC tool (Model for Urban Stormwater Improvement Conceptualization) which is a conceptual planning and design

tool that also contains a life cycle costing module for different NBS that are implemented in Australia (Jayasooriya et al., 2016; Khastagir and Jayasuriya, 2010; Schubert et al., 2017).

There are also other tools that can be used for modelling stormwater management options and/or to perform assessment of economic aspects of NBS in urban areas. These are documented in the work of Jayasooriya and Ng, (2014). However, most of these tools only focus on small-scale NBS such as bio-retentions, pervious pavements, green roofs, swales, constructed

wetlands, extended retention basins, retention ponds, sand filters, biofiltration tree planters and rainwater harvesting. There are only few tools that can address river and coastal flood protection measures and droughts while none of tools can be used to reduce the risk from landslides and storm surges. A lack of information systems, information clusters and platforms for exchange information between authorities and practitioners has been recognized by Kabisch et al., (2016).

There is also the need to explore the use of sensors, regulators, telemetry and Supervisory Control and Data Acquisition

(SCADA) systems for efficient and effective operation and real-time control of NBS. Such configuration, which is based on the use of real-time control technology for operation of NBS, can be referred to as "SMART NBS". The value of exploring SMART NBS configuration may be particularly beneficial for hybrid systems, where NBS sites need to be configured to work closely with different kinds of measures (e.g., traditional grey infrastructure measures).

**4.4 Socio-economic influence on implementation of NBS**

Investing in NBS for hydro-meteorological risk reduction is essential to ensure the capability for future socio-economic development (Faivre et al., 2018). In this respect, European Commission has been investing considerably in the research and innovation of NBS or EbA and some recent efforts are placed on practical demonstration of NBS for climate change adaptation and risk prevention (Faivre et al., 2017).

The European Commission is dedicated to bringing innovative 'sciences-policy-society' mechanisms, open consultations, and

knowledge-exchange platforms to engage society in improving the condition for implementation of NBS (Faivre et al., 2017). There are some inventories of web-portals, networks and initiatives that address NBS at European, national and sub-national levels (Table 4).

Denjean et al., (2017) noted that the people who propose NBS are in many cases ecologists and biologists who have been trained within a very different scientific paradigm and then speak a 'different language' than the key decision makers, who are

often civil and financial engineers, contractors and financing officers. Hence, this may limit the feasibility of implementation of NBS.





Very few articles study actions or processes in relation to stakeholder participation (Table 3). However, those that do so they stress the importance of involving stakeholders in the evaluation and implementation of NBS and the current practical limitations of implementing NBS. One of the important reasons for these is to ensure that stakeholders and local government are fully aware of multiple benefit of NBS so that they can integrate them better into planning for sustainable cities (Ishimatsu et al., 2017). For example, Liu and Jensen, (2018) and Chou, (2016) claim that the implementation of NBS with visible benefits on the landscape and the liveability of the city in terms of amenities, recreation, green growth, and microclimate can create positive attitudes among stakeholders towards applying NBS. Moreover, as the implementation of NBS is often a costly investment for local communities and the facilities are expected to be in place for a decade, it is essential to know the effectiveness of NBS (Semadeni-Davies et al., 2008). The involvement of researchers and stakeholders is important for monitoring, assessing and forecasting scenarios (Stanev et al., 2014). A case study of Great Plains in the US, Vogel et al., (2015) addressed how local perceptions of NBS effectiveness and applicability limit its adoption. One of the factors was a lack of awareness of NBS and support from stakeholders and authorities. Another case in Portland, Oregon, USA, Thorne et al., (2018) concluded that the limited adoption of NBS is caused by the lack of confidence in public preferences and socio-political structures as well as the uncertainty regarding scientific evidence related to physical processes. To solve this, they suggested that both socio-political and biophysical uncertainties must be identified and managed within the framework for designing and delivering sustainable urban flood risk management.

Schifman et al., (2017) proposed a Framework for Adaptive Socio-Hydrology (FrASH) that can be used in NBS planning and implementation by bringing ideas together from socio-hydrology, the capacity for adaptation, participation and inclusiveness, and organised action. The framework also helps in creating a connected network between municipalities, public works departments, organisations and people in the community. This potentially allows for the management of resilience in the system at multiple scales.

Often, it is not as easy to address socio-economic issues as technical questions. These socio-economic issues include perception and acceptance, policies, interdisciplinary nature of LID, education, and documenting the economic benefit of NBS implementation (Vogel et al., 2015). Nevertheless, qualitative research (i.e. surveys, interviews, and focus groups) helps to review and gain insights about the obstacles and motivations for implementing NBS as well as to understand a community's resilience and adaptive capacity (Matthews et al., 2015). For instance, bringing the findings to stakeholders and community members to discuss on what level of flood hazards is acceptable and what level of climate change adaptation capacity the community plans to achieve (Brown et al., 2012). Moreover, socio-political dynamics in NBS is still lacking, there are only few case studies available that critically evaluate the politics of NBS in the role of community mobilization (Triyanti and Chu, 2018).

Not only it is essential to involve stakeholders in the selection, planning, design and implementation of NBS, but it is also important for bridging gaps between researchers, engineers, politicians, managers and stakeholders. This may help to improve



our capacity for using both small and large scale NBS. There is well documented range of policy arrangements, scientific niches and current status of governance studies of NBS that was reviewed by Scarano, (2017) and Triyanti and Chu, (2018).

## 4.5 Multiple-benefits of NBS

The literature on NBS, SuDs, BMPS, LIDs, GI, EbA, and Eco-DRR increasingly refers to multiple benefits on social, economic
and environmental enhancements. The reason for that is that NBS are regarded as sustainable solutions that use ecosystem services to provide multiple benefits for human well-being and environment, which differ from grey infrastructure. One of the processes that could provide these benefits is to give more significant consideration to landscape and adaptive and multi-functionality design (Lennon et al., 2014; Vojinovic et al., 2017).

The literature to date shows that multiple challenges can be continually addressed through NBS. These include reducing flood
risk (Song et al., 2018), storing and infiltrating rainfall run-off, delaying and reducing surface runoff, reducing erosion and particulate transport (Loperfido et al., 2014) recharging groundwater discharge, reducing pollution from surface water (Donnell et al., 2018), increasing nutrient retention and removal (Loperfido et al., 2014), maintaining soil moisture, and enhancing vegetation growth.

Beyond water management, the case for these natural capital approaches includes their ability to provide additional benefits
on improving socio-economic aspects and human well-being through recreational areas and aesthetic value (Song et al., 2018), as well as encouraging tourism through the access to nature (Sutton-Grier et al., 2018). Green space can also provide a safe area for physical activity such as walking, jogging and cycling (Fan et al., 2011). Wheeler et al., (2010) quantified the volume and intensity of children's physical activity in greenspace and found that time in greenspace is more likely to lead to greater activity intensity amongst children. The use of NBS can bring economic benefits in different ways such as reduced/prevented
damage cost from hydro-meteorological events (Klijn et al., 2015), economic benefit from the reduction of stormwater that typically needs to be treated in a public sewerage system and energy and carbon savings from reduced building energy consumption (heating and cooling) (Soares et al., 2011).

The environmental benefits of NBS measures can have various positive impacts. Some of the most important are the ability to enhance environmental and ecosystem services by connecting habitat and biodiversity (Hoang et al., 2018; Reguero et al.,
2018; Thorslund et al., 2017b), increasing carbon consequences, reducing air and noise pollution (Donnell et al., 2018); and improving urban heat island effect mitigation (Raymond et al., 2017).

Zhang and Chui, (2019) reviewed the hydrological and bio-ecological benefits of NBS across spatial scales and suggested that there should be more research at the catchment scale to consider the full benefits of NBS. The hydrological and water quality benefits of NBS have been widely reviewed and discussed, but there are few articles that focus on the assessment of multi-
benefits of NBS. Hoang et al., (2018) proposed a new integrated methodology using a GIS approach to assess benefits and disadvantages of NBS, which include habitat connectivity, recreational accessibility, traffic movement, noise propagation,





carbon sequestration, pollutant trapping and water quality. Donnell et al., (2018) used BEST and the Blue-Green Cities toolbox to assess benefits, and Mills et al., (2016) assessed air pollution reduction based on tree canopy.

Alves et al., (2019) presented a novel methodology for valuing co-benefit for NBS application in urban contexts. In order to evaluate benefits effectively, Fenner, (2017) recommended that their spatial distribution should be assessed through multi-functional design making possible to identify how this is valuable to stakeholders and where the overall aggregated benefits occur. There is still a need for deeper understanding of assessment of multi-benefits in managing stormwater (Liu et al., 2017). A challenge is the lack of information on the values of ecosystem and multi-related ecosystems economic valuation (Bennett et al., 2009).

## 5 Conclusions

The present paper provides a critical review of the literature and identifies future research prospects based on the current knowledge gaps in the area of Nature-Based Solutions for hydro-meteorological risk reduction. The review process started by analysing 1381 articles sourced from Scopus from 2007 onwards. The articles sourced from Scopus were also cross-referenced with the articles from Web of Science and Google Scholar. The final full analysis was performed on 159 closely related articles.

The systematic review has shown that considerable achievements have been made to date. However, there are still many challenges and opportunities in extending the knowledge in NBS and that will play an important role in the coming years. Some of the key concluding remarks are summarised below.

There is a clear gap between the amount of research on small scale NBS in urban areas and large scale NBS at the catchment (river basin), rural, and regional scale. The reason for this is that a large-scale system is more complex than a small system. Therefore, the research and frameworks that deal with the problem of reducing hydro-meteorological risk with upscaling NBS from urban scale to catchment (river basin) scale would be beneficial, and it would be also beneficial to understand both the natural processes of large scale NBS and how they change over time. Furthermore, there are only few studies that combine NBS at both small- and large-scale and further research in this direction is highly desirable.

Obviously, there is no single NBS solution that can solve all problems. Every project needs to be designed to address a particular challenge in its local contexts and in its respective community. Therefore, an understanding of site conditions is necessary for NBS to achieve the target of the project.

Based on the findings of the literature review, there are still challenges in relation to methods and tools for planning and implementing NBS. These include improving and developing methods for assessing co-benefits (especially socio and ecological benefits i.e. aesthetics values, community livability, and human health), frameworks and methods for evaluating large-scale NBS and "hybrid measures" (i.e. combinations of grey infrastructure and small and large scale NBS).



There are also challenges in incorporating local stakeholder participation within the framework and models and within the assessment and implementation process. Other challenges regarding governance are to develope guidance on effective models of governance, provide insights information on actors, institutions and legal instruments and other requirements that are relevant for implementing NBS. The reason for this is the lack of workable frameworks that can bring together variety of

stakeholder groups. Moreover, there is still a lack of finance studies and guidelines for cost-effective implementation, maintenance and operation of NBS projects and mechanisms that can be used to promote new business and finance models for successful implementation of NBS.

There should be also more efforts in the development of assessment tools that incorporate new technologies such as real-time control systems, forecast models, and coupled models to provide more active and integrated operational solutions (i.e., SMART

NBS). There is the need for the development of databases that include functions, benefits, and costs of large and small scale NBS to facilitate future research.

Overall, investments in NBS will benefit society by providing cost-effective measures and adaptive strategies that protect their communities and achieve a range of co-benefits. Therefore, bridging the gaps between researchers, engineers and stakeholders will help to improve the capacity of NBS in reducing hydro-meteorological risk as well as considering the multitude of

opportunities and benefits of NBS for co-creation and co-development in intensive participation process. Strengthening this aspect maybe beneficial in improving acceptance at local level.

## 6 Acknowledgements

Production of this article received funding from the European Union's Horizon 2020 Research and Innovation programme under grant agreement No 776866 for the research RECONECT (Regenerating ECOsystems with Nature-based solutions for

hydro-meteorological risk rEduCTion) project. It was also supported by the European Union's Horizon 2020 Research and Innovation programme under grant agreement No 776848 for OPERANDUM and under grant agreement No 776681 for PHUSICOS. The study reflects only the authors' view and the European Union is not liable for any use that may be made of the information contained herein.

## Appendix

## Appendix A: Abbreviations

| | |
|---|---|
| AMS | Adaptive metropolis search |
| AST | Adaptation Support Tool |
| BeST | Benefits of SuDS Tool |
| BGI | Blue-Green Infrastructure |





| | |
|---|---|
| BMPDSS | Best Management Practice Decision Support |
| BMPs | Best Management Practices |
| CBA | Cost-benefit analyses |
| CBD | Convention on Biological Diversity |
| 5 CCA | Climate change adaptation |
| CEM | Commission on Ecosystem Management |
| DE | Differential evolution |
| DRR | Disaster risk reduction |
| EbA | Ecosystem-based Adaptation |
| 10 Eco-DRR | Ecosystem-based Disaster Risk Reduction |
| EC | European Commission |
| FrASH | Framework for Adaptive Socio-Hydrology |
| GI | Green Infrastructure |
| IIED | International Institute for Environment and Development |
| 15 IUCN | International Union for Conservation of Nature |
| LCC | Life cycle costing |
| LID | Low Impact Development |
| MAUT | Multiattribute utility theory |
| MCA | Multi-criteria analysis |
| 20 MLSOP | Multilevel spatial optimization |
| MOEA | Most popular multiobjective evolutionary algorithms |
| MOO | Multi-Objective Optimal |
| MOPSO | Multi-objective particle swarm optimisation |
| MOUSE | Model of Urban Sewers |
| 25 MUSIC | Model for Urban Stormwater Improvement Conceptualization |
| NBS | Nature-Based Solutions |
| NSGA-II | Non-dominated Sorting Genetic Algorithm II |
| PSO | Particle swarm optimisation |
| RECONECT | Regenerating ECOsystems with Nature-based solutions for hydro-meteorological risk rEduCTion |
| 30 ROI | Return on Investment |
| RPE | Relative Performance Evaluation |
| SA | Simulated Annealing |
| SCADA | Supervisory Control and Data Acquisition |



| SCP | Sponge City Programme |
|---|---|
| SDGs | Sustainable Development Goals |
| SEI | Stockholm Environment Institute |
| SFDRR | Sendai Framework for Disaster Risk reduction |
| SuDS | Sustainable Urban Drainage Systems |
| SUSTAIN | System for Urban Stormwater Treatment and Analysis IntegratioN |
| SWAT | Soil and Water Assessment |
| SWIMM | Storm Water Management Model |
| TSS | Total Suspended Solids |
| UN | United Nations |
| UNFCCC | UN Framework Convention on Climate Change |
| US EPA | United States Environmental Protection Agency |
| UWOT | Urban Water Optioneering Tool |
| WCPA | World Commission on Protected Areas |
| WSUD | Water Sensitive Urban Design |

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



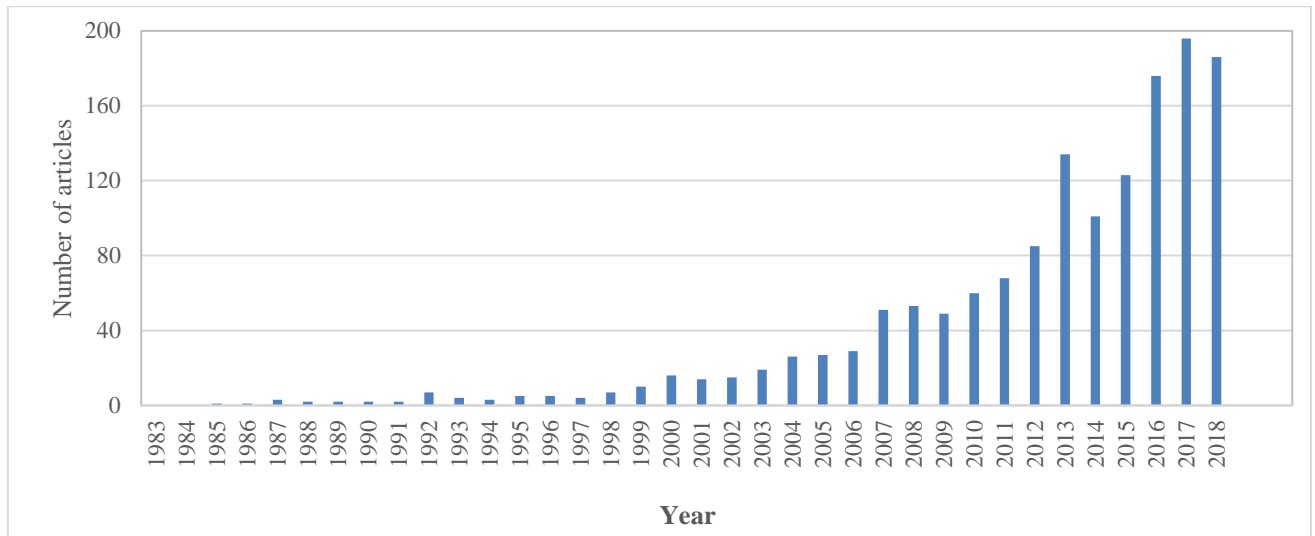

**Figure 1: Number of articles per year on Nature Based Solutions for hydro-meteorological risk reduction sourced from Scopus over the period 1983-2018.**

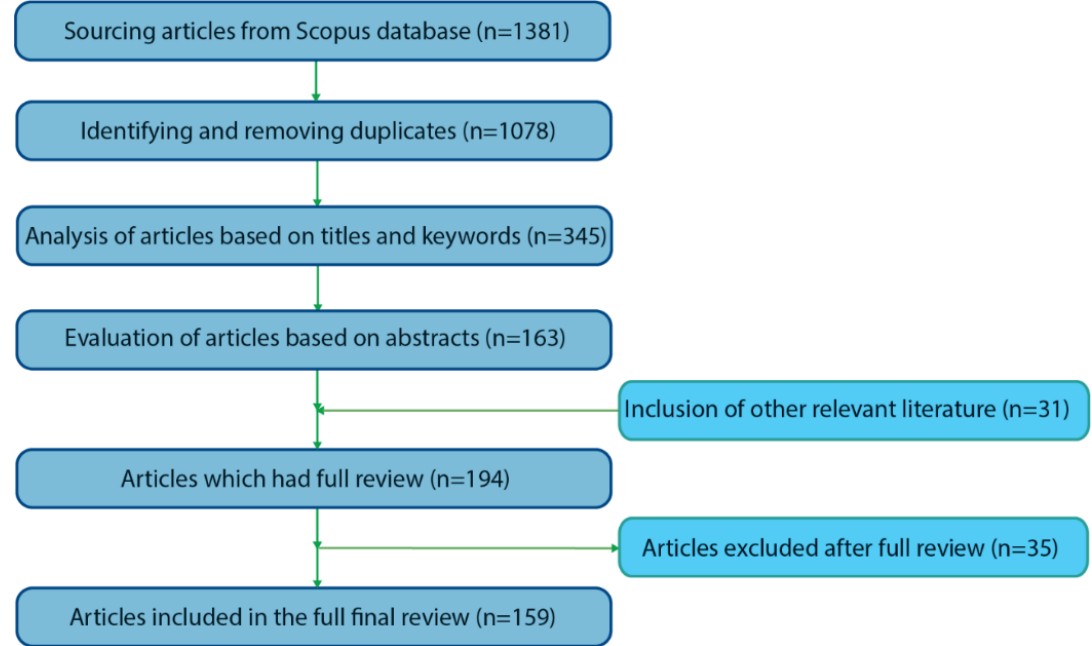

5    **Figure 2: Process of article selection on Nature Based Solutions for hydro-meteorological risk reduction. The process started with sorting 1381 articles and 31 other documents. The final number of fully reviewed articles is 159.**





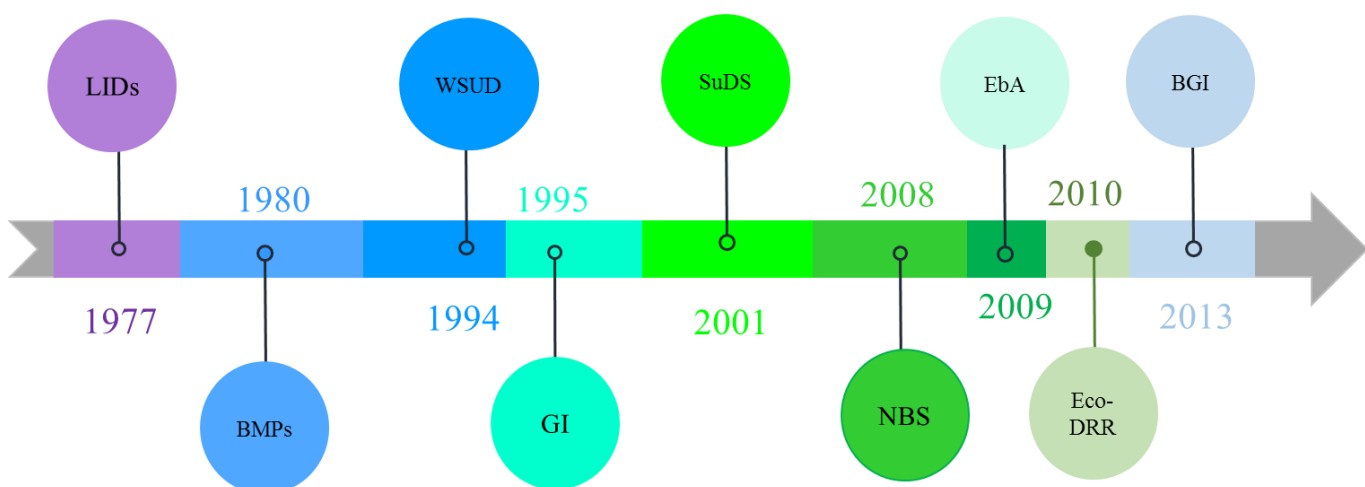

**Figure 3: Timeline/year of origin of each terminology (Low Impact Developments (LIDs), Best Management Practices (BMPs), Water Sensitive Urban Design (WSUD), Green Infrastructure (GI), Sustainable Urban Drainage Systems (SuDS), Nature-Based Solitions (NBS), Ecosystem-based Adaptation (EbA), Ecosystem-based Disaster Risk Reduction (Eco-DRR) and Blue-Green Infrastructure (BGI)) based on their appearance in publications**

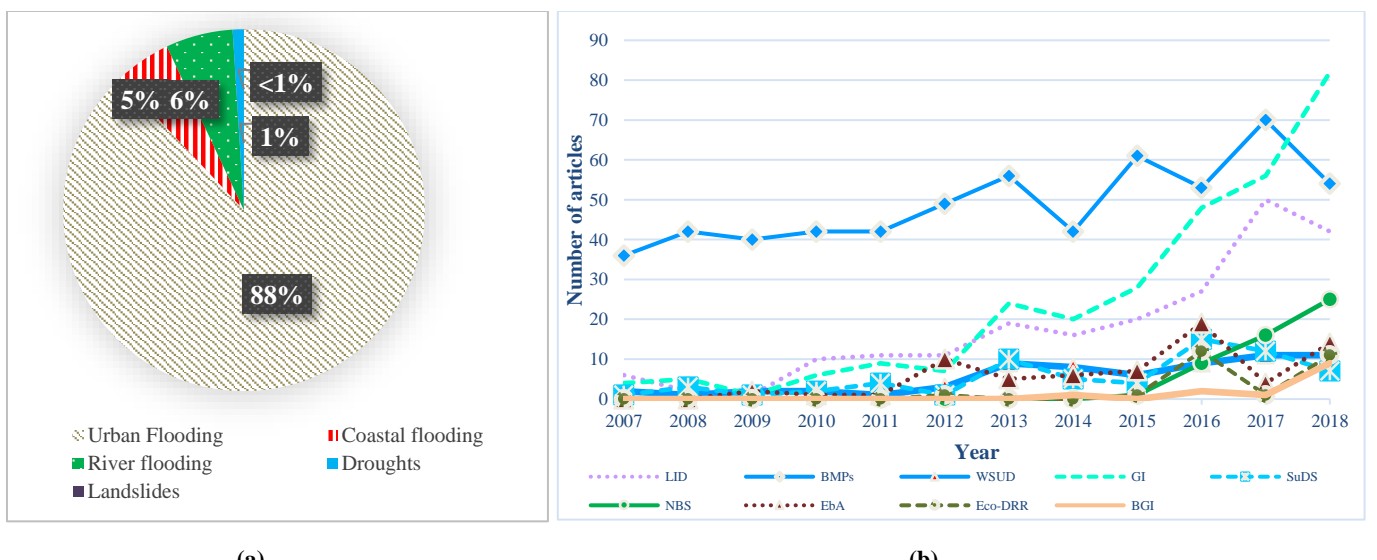

| (a) | (b) |
| --- | --- |

**Figure 4: An overview of published articles on Nature-Based Solutions for hydro-meteorological risk reduction: (a) percentage of published articles that have been studied for reducing urban flooding, coastal flooding, river flooding, droughts and landslides and (b) number/trend of published articles for Low Impact Developments (LIDs), Best Management Practices (BMPs), Water Sensitive Urban Design (WSUD), Green Infrastructure (GI), Sustainable Urban Drainage Systems (SuDS), Nature-Based Solitions (NBS), Ecosystem-based Adaptation (EbA), Ecosystem-based Disaster Risk Reduction (Eco-DRR) and Blue-Green Infrastructure (BGI)**



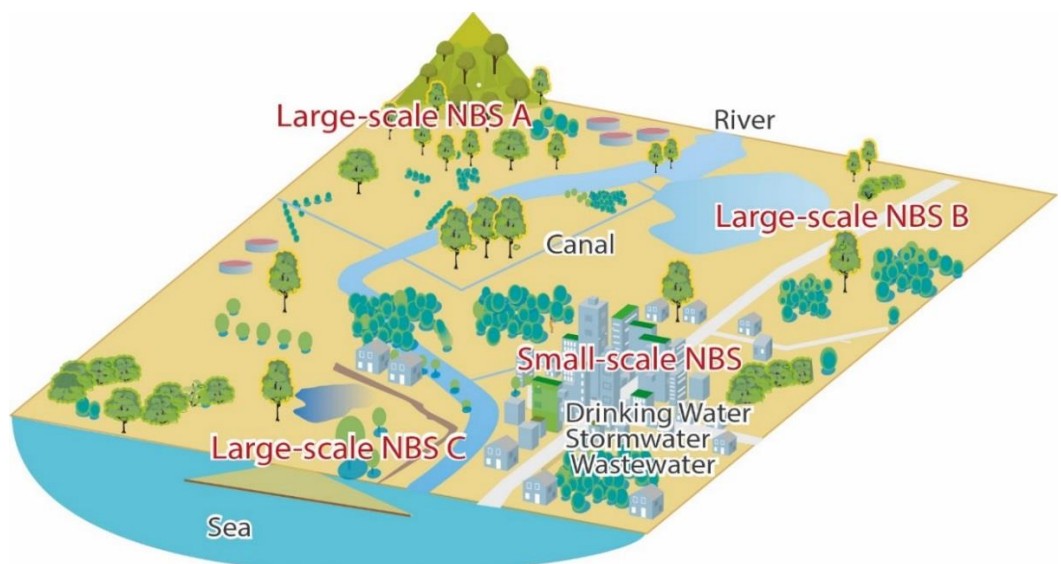

**Figure 5: Illustration of large and small scale Nature-Based-Solutions (NBS); Large-scale NBS A illustrates NBS in mountainous regions (e.g., afforestation, reforestation, slope stabilization, etc.), Large-scale NBS B illustrates NBS along river corridors (e.g., dike relocation, retention basins, etc.) and Large-scale NBS C illustrates NBS in coastal regions (e.g., sand dunes, protection dikes/walls, marshes, etc.); Typical examples of Small-scale NBS are green roofs, green walls, rain gardens, porous/permeable pavements, swales, bio-retention, etc.**

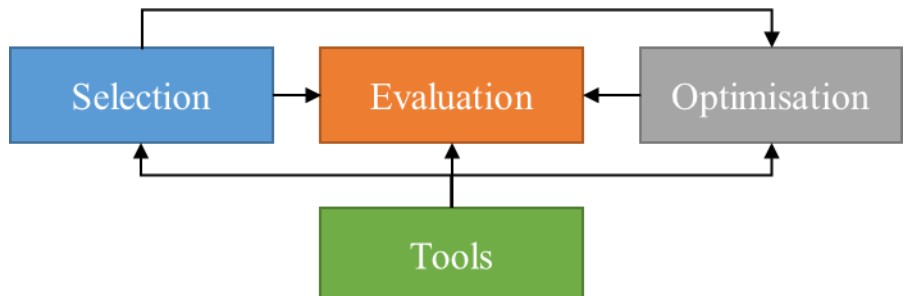

**Figure 6: Evaluation process of Nature-Based Solutions**

**Table 1: Selected concepts and terms used to search relevant literature on NBS for hydro-meteorological risk reduction**

| No | Research words | | |
|---|---|---|---|
| | **First concept**<br>**(Nature-Based Solutions)** | **Connection** | **Second concept**<br>**(Hydro-meteorological risk)** |
| 1 | "Nature-based solutions" OR | AND | "Flood" |
| 2 | "Nature-Based Solutions" OR | AND | "Drought" |
| 3 | "Low impact development" OR | AND | "Storm surge" |
| 4 | "Sustainable Urban Drainage Systems" OR | AND | "Landslide" |
| 5 | "Water Sensitive Urban Design" OR | AND | "Hydro-meteorological" |
| 6 | "Best Management Practices" OR | AND | "Disaster" |
| 7 | "Green infrastructure" OR | AND | "Review" |
| 8 | "Green blue infrastructure" OR | AND | "Hydrology" |
| 9 | "Ecosystem-based Adaptation " OR | AND | "Coastal" |
| 10 | "Ecosystem-based disaster risk reduction " | | |





**Table 2: Glossary of terminologies and their geographical usage**

| Terminology | Definition/Objectives/Purpose | Commonly used in | Reference |
|---|---|---|---|
| Low Impact Development (LIDs) | *"LID is used as a retro-fit designed to reduce the stress on urban stormwater infrastructure and/or create the resiliency to adapt to climate changes, LID relies heavily on infiltration and evapotranspiration and attempts to incorporate natural features into design."* | - United States<br>- New Zealand | (Barlow et al., 1977; County, 1999; Eckart et al., 2017) |
| Best management practices (BMPs) | *"A device, practice or method for removing, reducing, retarding or preventing targeted stormwater runoff constituents, pollutants and contaminants from reaching receiving waters"* | - United States<br>- Canada | (Biggers et al., 1980; Moura et al., 2016; Strecker et al., 2001) |
| Water Sensitive Urban Design (WSUD) | *"Manage the water balance, maintain and where possible enhance water quality, encourage water conservation and maintain water-related environmental and recreational opportunities".* | - Australia | (Lottering et al., 2015; Mouritz, 1996; Whelans consultants et al., 1994) |
| Sustainable Urban Drainage Systems (SuDS) | *"Replicate the natural drainage processes of an area – typically through the use of vegetation-based interventions such as swales, water gardens and green roofs, which increase localised infiltration, attenuation and/or detention of stormwater"* | - United Kingdom | (Abbott and Comino-Mateos, 2001; Ossa-Moreno et al., 2017) |
| Green Infrastructure (GI) | *"The network of natural and semi-natural areas, features and green spaces in rural and urban, and terrestrial, freshwater, coastal and marine areas, which together enhance ecosystem health and resilience, contribute to biodiversity conservation and benefit human populations through the maintenance and enhancement of ecosystem services"* | - United states<br>- United Kingdom | (Gill et al., 2007; Lafortezza et al., 2013; Naumann et al., 2011; Walmsley, 1995) |
| Ecosystem-based Adaptation (EbA) | *"The use of biodiversity and ecosystem services as part of an overall adaptation strategy to help people to adapt to the adverse effects of climate change."* | - Canada<br>- Europe | (CBD, 2009; McVittie et al., 2017; Scarano, 2017) |
| Ecosystem-based disaster risk reduction (Eco-DRR) | *"The sustainable management, conservation, and restoration of ecosystems to reduce disaster risk, with the aim of achieving sustainable and resilient development"* | - Europe<br>- United states | (Estrella and Saalismaa, 2013; PEDRR, 2010; Renaud et al., 2016) |
| Blue-Green Infrastructure (BGI) | *"BGI provides a range of services that include; water supply, climate regulation, pollution control and hazard regulation (blue services/goods), crops, food and timber, wild species diversity, detoxification, cultural services (physical health, aesthetics, spiritual), plus abilities to adapt to and mitigate climate change"* | - United Kingdom | (Bozovic et al., 2017; Lawson et al., 2014; PEDRR, 2010; Rozos et al., 2013) |
| Nature-Based Solution | *"NBS aim to help societies address a variety of environmental, social and economic challenges in sustainable ways. They are actions inspired by, supported by or copied from nature; both using and enhancing existing solutions to challenges, as well as exploring more novel solutions."* | - Europe | (Cohen-Shacham et al., 2016; European Commission (EC), 2015; Faivre et al., 2017; MacKinnon et al., 2008; Stürck et al., 2015) |





**Table 3: Overview of knowledge gaps and potential future research prospects**

| Subject | Number of publications | Knowledge Gaps | Future research prospects |
|---|---|---|---|
| **1. The effectiveness of small scale NBS** | 39 | - Combination of small and large scale NBS with grey infrastructure. | • Development of a framework and methods to upscale NBS from small to large scale.<br>• Development of a framework, methods and tools to select, evaluate, and design hybrid measures for hydro-meteorological risk reduction |
| | | - NBS for droughts, landslides and storm surges. | • Application of NBS to reduce the risk of droughts, landslides and storm surges. |
| **2. The effectiveness of large scale NBS** | **7** | - Application to hydro-meteorological risk reduction;<br>- Combination of large scale NBS with grey infrastructure | • Development of a framework, methods and tools to select, evaluate, and design large scale NBS individually and in hybrid combinations for hydro-meteorological risk reduction<br>• Development of typologies and guidelines for NBS design, implementation, operation and maintenance. |
| **3. Selection and assessment of NBS with the focus on risk reduction** | 26 | Framework for selection of NBS | • Defining the role of ecosystems in terms of risk reduction, socio-economic and hydro-geomorphological settings<br>• Combining spatial planning and stakeholders participation in the co-selection process |
| | | Framework for cost analysis | • Combining economic value of ecological damage and environmental impact, including the "invisible" ecosystem services (see also Estrella et al., 2013)<br>• Application of the whole life cycle costing and return on investment within the cost-benefit analysis of NBS<br>• Comparing costs and benefits between NBS, GI and hybrid measures<br>• Defining opportunity costs and trade-offs of NBS implementation |
| | | Framework for optimal configuration of NBS | • Use of optimisation techniques to maximise the main benefit and co-benefits of NBS while minimising their costs.<br>• Use of optimisation techniques to maximise the efficiency of NBS and to define their best configurations within hybrid solutions.<br>• Assessing the effectiveness of solutions on short and long terms |
| | | Combination between multi-criteria and qualitative research | • Use of multi-criteria and qualitative research in evaluation of NBS.<br>• How to combine quantitative and qualitative data and research methods.<br>• Application of qualitative research methods and interviews to effectiveness of NBS |
| **4. Multi-benefits of NBS** | 21 | Assessment of multi-benefits of NBS | • Quantification of co-benefits.<br>• Development of a framework, methods and tools to evaluate wide ranging intangible and tangible benefits.<br>• Gaining deeper understanding of NBS benefits for human well-being |
| | | Assessment of ecosystem capacity | • Assessing ecosystem capacity to maintain services over a longer period of time (see Estrella and Saalismaa, 2013)<br>• Long–term monitoring and evaluation of ecosystem performance and function before and after the disaster<br>• Addressing the complexity of coupled social and ecological systems |





**Table 3: Overview of knowledge gaps and potential future research prospects (continue)**

| Subject | Number of publications | Knowledge Gaps | Future research prospects |
|---|---|---|---|
| **5. Application of tools** | 18 | Application of new technologies and concepts (e.g., high resolutions numerical models, complex, crowdsourcing tools, real-time control system) | • Integration of real-time monitoring and control technologies for NBS operation.<br>• A trade-off between high resolution numerical models and accuracy of results.<br>• Use of novel modelling techniques such as complex adaptive systems models and serious games. |
| | | Web-based decision support tools/systems | • Development of databases of small and large scale NBS for hydro-meteorological risk reduction.<br>• Development of platforms, info-systems and clusters for exchange knowledge (see also Kabisch et al., 2016).<br>• Development of tools to support decision makers in selecting and evaluating hybrid measures.<br>• Development of tools to assess the multiple-benefits for small and large scale NBS and their hybrid combinations. |
| **6. Multifunctional design** | 2 | Framework for multifunctional design | • Development of a framework and methods to support multifunctional design.<br>• Application of novel landscape design techniques.<br>• Combining the knowledge from landscape architecture and water engineering (Kabisch et al., 2016). |
| **7. Stakeholders participation** | 8 | Frameworks for effective stakeholder involvement and co-creation | • Frameworks for involvement of stakeholders in the selection, evaluation, design, implementation, and monitoring of NBS (i.e., the co-called co-creation process). |
| **8. Financing, governance and policy** | 4 | Desirable governance structures to support effective implementation and operation of NBS at different scales and contexts | • Information concerning legal instruments and requirements.<br>• Development of effective governance structures<br>• Compilation of data and information concerning multiple actors and institutions which are relevant for implementation of NBS<br>• Understanding water governance structures, drivers, barriers and mechanism for enabling system transformation (see also Albert et al., 2019)<br>• Development of methods for evaluation of social, political and institutional dimensions of NBS (see also Triyanti and Chu, 2018) |
| | | Desirable finance models (e.g., public-private partnerships, blended financing, etc.) | • Development of finance guidance for implementing maintaining and operating NBS projects<br>• Guidelines concerning development of new business and finance models (see also Kabisch et al., 2016)<br>• Development of financial mechanisms to engage public and private sectors in the implementation of NBS |
| | | Bridging gaps between science-practice-policy | • Bridging gaps between researchers, engineers, authorities and local stakeholders.<br>• Bridging the policy and institutional gaps.<br>• Bringing innovation to engage society in implementing and improving NBS. |



**Table 4: An overview of web-portals, networks and initiatives that address Nature-Based Solutions**

| Name | References/ Website | Terminology used | Scale level | Funded by | Proposes |
|---|---|---|---|---|---|
| **OPPLA** | (Oppla, 2019) | Nature-Based Solution, Natural capital, Ecosystem services | Europe | FP7 (EC) | A new knowledge marketplace - EU repository of NBS; a place where the latest thinking on ecosystem services, natural capital and nature-based solutions is brought together. |
| **BiodivERsA** | (Biodivera, 2019) | Ecosystem services | Europe | Horizon 2020 (EC) | A network of funding organizations promoting research on biodiversity and ecosystem services. |
| **BISE** | (BISE, 2019) | Ecosystem services, Green infrastructures | Europe | EC | A single entry point for data and information on biodiversity supporting the implementation of the EU strategy and the Aichi targets in Europe. |
| **ThinkNature** | (ThinkNature, 2019) | Nature-Based Solution | Europe | Horizon 2020 (EC) | A multi-stakeholder communication platform that supports dialog and understanding of NBS. |
| **ClimateADAPT** | (Climate ADAPT, 2019) | EbA, Nature-Based Solution, GI | Europe | EC, EEA | A platform that supports Europe in adapting to climate change by helping users to access and share data and information relevant for CCIVA. |
| **Natural Water Retention Measures** | (NWRM, 2019) | Natural water retention measures | Europe | EC | A platform that gathers information on NWRM at EU level. |
| **Urban Nature Atlas** | (NATURVATION, 2019) | Nature-Based Solution | Europe | Horizon 2020 (EC ) | A platform that contains around 1000 examples of Nature-Based Solutions from across 100 European cities. |
| **Disaster Risk Management Knowledge Centre** | (DRMKC, 2019) | Eco-DRR | Europe | EC | A platform that provides a networked approach to the science-policy interface in DRM. |
| **Natural Hazards – Nature Based Solutions** | (World Bank et al., 2019) | Nature-Based Solution | Global | The World Bank | A project map that provides a list of nature-based projects that are sortable by implementing organisation, targeted hazard, and type of nature-based solution, geographic location, cost, benefits, and more. |
| **Nature-based Solutions Initiative** | (Nature-based Solutions Initiative, 2019) | Nature-Based Solution | Global | International Institute for Environment and Development (IIED) | The global policy platform that provides information about climate change adaptation planning across the globe openly available and easy to explore. |
| **weADAPT** | (SEI, 2019) | Ecosystem-based Adaptation | Global | Stockholm Environment Institute (SEI) | A collaborative platform on climate adaptation issues, which allows practitioners, researchers and policy-makers to access credible, high-quality information and connect. |
| **Nature of Cities** | (The Nature of Cities, 2019) | Green Infrastructures | Global | | An international platform for transdisciplinary dialogue concerning urban solutions. |
| **ClimateScan** | (ClimateScan, 2019) | Blue-Green Infrastructures | Global | EC | Global online tool which acts as a guide for projects and initiatives on urban resilience, climate proofing and climate adaptation around the world. |