# Peer review of "Nature-Based Solutions for hydro-meteorological risk reduction: A state-of-the-art review of the research area"

_Natural Hazards and Earth System Sciences, 2019_

## Referee Comment (RC1) · Anonymous Referee #1 · 22 Jun 2019

Dear editor,dear authors,

The premise of this article is extremely interesting and some of the conclusions of the article, in particular the "Overview of knowledge gaps / potential future research" is a very useful contribution to advancing this topic. The article helps to give an overview of the many concepts and terms associated with Nature based solutions for disaster risk reduction and it attempts to provide a mixed quantitative /qualitative assessment of a number of pre-determined questions that the authors have outlined as the objectives of the review.

It therefore merits to be published if some fundamental methodological issues can be

resolved.

1. Concepts The article provides an interesting historical overview of the different related concepts but there is still a confusion of terms. The abstract in particular is confusing, i.e. Nature based Solutions (NbS) is generally considered to be an umbrella term under which other types of approaches, Eba, Eco-DRR and GI / GBI provide more specific solutions to more specific issues (see various definitions given by IUCN and EU-related). This does not come out clearly in the article.

For example: p. 4/ line 30 NbS is not just about storm water

2. Methodology of the review This is where this reviewer has the greatest number of questions: - Good that multiple data bases were used but why assume that just because Scopus has the greatest number of articles, that it is the most comprehensive? You could have merged all three searches and then removed duplicates. - Adding missing articles adds a huge bias to your search. Which articles were selected and based on what criteria? That the keywords were there? - Which criteria were used for deleted certain articles - perhaps I missed this? - Search terms: you had several search terms from your first column with "urban", this may have included a bias toward urban - One of the main objectives of this review was to find trends and patterns, so only section 4.1 Trends, knowledge gaps and future research prospects provides quantitative results, the remaining sections onward are mainly qualitative descriptions to answer your pre-defined research questions: e.g. (2) Effectiveness of multiple NBS sites, etc. It should be clarified that you the review is quantitative but also qualitative based on pre-defined questions. - However you do not justify why you selected these topics - again, they did not emerge as trends in the literature, you selected them and then found literature to analyse them . In other words, you combine deductive with inductive research. This should be made more explicit, or you should chose one or the other.

3. Other - Some paragraphs appeared to be more a promotion of author's projects

rather than related to the literature review ?? They might belong in the conclusions but not as part of the analysis.. -The manuscript needs to be redrafted by a native English speaker. e.g. p8, line 27 "desiderative" ;) - The table on websites related to the topic is good but excludes a few important sites, namely IUCN's data base on EbA projects and the Partnership for Environment and Disaster Risk Reduction (PEDRR) website

This reviewer hopes the above comments can be taken into consideration as this work deserves to be published.

---

## Referee Comment (RC2) · Anonymous Referee #2 · 15 Jul 2019

Nature-Based Solutions for hydro-meteorological risk reduction: A state-of-the-art review of the research area

by Laddaporn Ruangpan, Zoran Vojinovic, Silvana Di Sabatino, Laura Sandra Leo, Vittoria Capobianco, Amy M. P. Oen, Michael McClain, Elena Lopez-Gunn

Comments to the Author Summary of the manuscript This manuscript (ms) reviews scientific publication on Nature-based solutions (NBS) for hydro-meteorological risk reduction and related terms. The authors proceeded in a systematic way by using search terms in various scientific literature databases and analyzed over 1000 references. The ms concludes by summarizing the main findings and suggesting further

research in some of the reviewed areas.

Evaluation I think the topic of this manuscript is highly relevant and important in order to review NBS to tackle the ecological crisis the world is facing. Accordingly, I do think that this ms should be considered for publication. However, I have my major doubts if the presented ms really helps to summarize the vast amount of literature on NBS and if it really identifies the knowledge gap in order to be able to recommend the area of focus for future research. My main concerns are the following:

i) Methodology: a simple search for "Nature-based solutions" in the WoS shows that three of the four most relevant and most cited papers have not been considered in this ms (Keesstra et al. 2018, Nesshover et al. 2017, and Eggermont et al. 2015). Accordingly, I would recommend revising the method of selecting research articles that are being taken into account in the review. ii) Structure: I recommend limiting the structure to three levels of subsection: especially section 4 could be better structured, avoiding sections with titles that do not clearly adhere to a three-level subsection structure. iii) Content is more valuable than academic metrics: while I do see a value in using academic metrics and search engines to select relevant literature, it would be helpful to review the actual characteristics, benefits, and scales of various NBS. Specifically, it would be helpful to have a table that summarizes area, volume of water retention, costs, and effectiveness (and other characteristics) of different NBS. The number of articles does not indicate anything about the effectiveness of a NBS, accordingly, I would encourage the authors to focus more on the characteristics of NBS rather than the number of articles found. In short, more quantitative assessments of the benefits of NBS rather than generic statements would be highly appreciated. iv) Definitions: in my opinion, it would be helpful to provide a table with definitions and examples of the various academic terms used in the review: The study provides generic definitions for GI, EbA, and NBS, but it is left upon the reader to interpret the definitions. I would recommend to complement Table 2 with some quantitative figures on water retention, area, costs, advantages, disadvantages etc.... (see also the previous comment). v)

Drought: it is well know that land reclamation and restoration reduces evaporation and mitigates the drought risk. However, the authors found only one single study referring to the drought risk. This might be due to a methodology based on "key words" rather than content. vi) Scale and examples: one example that struck me is the NBS "Room for the River Programme" in the Netherlands at the Rhine and Meuse. It is general knowledge that flood protection has to start upstream in the headwaters, where most of the precipitation occurs, to be efficient. Nevertheless, the ms only mentions NBS in the Netherlands (a third of the Netherlands are below sea level and sea levels are rising), ignoring the far more relevant NBS in upstream countries. This might be linked to the somewhat limited methodology of the literature review (see comment i). vii) Tools: in my opinion, the review of tools could be shortened, as it is slightly off the topic. Instead, more attention could be given to the quantification of the various benefits of NBS could be provided (see comment iii). viii) Conclusion: the current conclusion provides general and generic statements and any reader somewhat familiar with the topic does not really learn anything new. It would be helpful to generate more conclusive and quantitative statement based on the review: which NBS are most effective, which provide most multi-benefits, which require least areas, which are most accepted?

I recommend that the authors revise this ms thoroughly and resubmit it again for publication.

---

## Author Comment (AC1) · 20 Aug 2019

**Reponses to first referee's comments on "Nature-Based Solutions for hydro-meteorological risk reduction: A state-of-the-art review of the research area" by Laddaporn Ruangpan et al.**

The premise of this article is extremely interesting and some of the conclusions of the article, in particular the "Overview of knowledge gaps / potential future research" is a very useful contribution to advancing this topic. The article helps to give an overview of the many concepts and terms associated with Nature based solutions for disaster risk reduction and it attempts to provide a mixed quantitative /qualitative assessment of a number of pre-determined questions that the authors have outlined as the objectives of the review. It therefore merits to be published if some fundamental methodological issues can be resolved.

**Authors' response:** Thank you for your encouragement and comments. Your concerns are addressed in this response letter and the manuscript revised accordingly. Please find our point-by point response below.

**1. Concepts**

**Comments from Referee:** The article provides an interesting historical overview of the different related concepts but there is still a confusion of terms. The abstract in particular is confusing, i.e. Nature based Solutions (NbS) is generally considered to be an umbrella term under which other types of approaches, Eba, Eco-DRR and GI / GBI provide more specific solutions to more specific issues (see various definitions given by IUCN and EU-related). This does not come out clearly in the article.

For example: p. 4/ line 30 NbS is not just about storm water

**Authors' response:** Thank you for pointing out this issue. We agree that terminology was confusing in the Abstract and other instances. This has been clarified in the revised manuscript. Furthermore, Section 3 "Overview of definitions and theoretical backgrounds", has been modified and expanded to better highlight the definition of NBS as an umbrella concept, as the reviewer suggested. This section also has been relocated to section 2 before "Materials and methodology" section as it discusses more on the background of NBS.

P.4 line 30: revised. Now, we specifically refer to SuDs, LIDs and WSUD terms in the sentence.

**Authors' change in the revised manuscript** (revised and added text with yellow highlights)**:**

[revised manuscript text omitted]

**2. Methodology of the review**; This is where this reviewer has the greatest number of questions:

**Comments from Referees 2.1** Good that multiple data bases were used but why assume that just because Scopus has the greatest number of articles, that it is the most comprehensive? You could have merged all three searches and then removed duplicates.

**Authors' response 2.1** Thank you very much for your comment. The authors have revised the methodology (see also next comment) by including both Web of Science and Scopus databases and merged the two searches together as recommended by the reviewer, and removed duplicates. Note that Google Scholar has been completely excluded from the revised methodology because it has limited metadata and filters which, at present, do not allow to limit results to articles published in peer-reviewed, scientific journals written in English (one of the three selection criteria adopted in our search process).

**Comments from Referees 2.2** Adding missing articles adds a huge bias to your search. Which articles were selected and based on what criteria? That the keywords were there? - Which criteria were used for deleted certain articles - perhaps I missed this?

**Authors' response 2.2** We agree that the methodology of this review was not clearly explained and had some flaws. Thanks to Reviewer's comments, our methodological approach has been carefully revised and improved. Specifically:

1) Bias introduced by missing articles has been removed, namely those articles are no longer evaluated neither included/added in the analysis. Note that few comments drawn upon this subset of articles have been retained because considered of relevance to our discussion, but they are now included in the new Section 2, which is not part of the "Findings" section

2) An analysis of why other papers in the extended list did not appear in the search shows that they were missed because they use the terms 'green and grey infrastructure' as

opposed to 'green infrastructure' directly. As this is merely a language issue, the term 'green and grey infrastructure' was added to the search terms.

1) As this Reviewer pointed out, the selection process was not clearly explained in the original manuscript. We have now substantially expanded the methodological section, by explicitly stating the objectives of the review and by explaining the criteria used for selecting the literature of relevance with respect to these objectives. This is summarized in the diagram below (included in the new version of the manuscript) which shows that the method consists of two phases. For the search process (phase I) the only selection criteria adopted were that (a) articles are published in peer-reviewed and scientific journals written in English; (b) articles reported on NBS in terms of hydro-meteorological risk reduction (construction of the search query based on the keywords in Table 1); (c) articles were published in the period 2007 to 1 December 2018. The search process resulted in a total of 1204 articles which were then subjected to selection process (Phase II). The selection process involved a set of progressive steps as schematized in Fig.3 and detailed in the following: << *Initially, all articles were analysed on the basis of reading titles and keywords and their relation to the search terms. For example, articles having 'resilience', 'stormwater' or other relevant words in the title or keywords were selected for continued analysis. Secondly, a more in-depth analysis was conducted, based on reading the abstract of each article selected in the previous step. The criteria was that the abstract should discuss about hydro-meteorological risk reduction. For example, if the abstract of the articles focuses more on water quality than risk, then that paper was excluded. This step served to reduce the number of articles from 380 to 185. Finally, reading full articles was undertaken to identify those that were relevant to the review objectives. Any studies appearing to meet the key objectives (dealing with subjects such as effectiveness of NBS, techniques, method and tools for planning, and others which are of relevance for the key objectives) would then be included in the review. As a result, the entire selection process resulted in a total of 137 articles were selected* >> (text extrapolated from the revised Section 2.2 (now Section 3.2)). For sake of completeness and clarity, the new version of the entire methodological section is provided below.

**Authors' change in the revised manuscript:**

**3. Materials and methodology**

Explain here that the entire methodology consisted of two phases as schematized in the diagram (Fig.3). The first phase consisted in the identification of all articles satisfying the searching criteria discussed in Section 3.1 Next, all articles were screened and filtered in or out based on the selection criteria discussed in section 3.2.

[Figure]

**Figure 3: Process of article selection on Nature Based Solutions for hydro-meteorological risk reduction. The final number of fully reviewed articles is 137.**

**3.1 Search strategy**

The review analysis concerned articles in peer review and scientific journals written in English. Two main concepts were used in the search: Nature-Based Solutions and hydro-meteorological risk. As the concept of 'Nature-Based Solutions' appears under different names (which more or less relate to the same field of research), articles related LIDs, BMPs, WSUD, SuDS, GI, BGI, EbA and Eco-DRR were included in the identification of relevant articles (see Table 2). The review of hydro-meteorological risk included literature on relevant terms (i.e. disaster, review etc.) and different types of risk (i.e. floods, droughts, storm surges and landslides) (Table 2).

During the construction of the queries, the strings were searched only within Index terms and Metadata "titles, abstract, and keywords" in the Scopus and Web of Science database. The search terms for the two concepts were linked with the Boolean operator "AND" while the Boolean operator "OR" was used to link between the possible terms (Table2). An example of a protocol is shown below:

"TITLE-ABS-KEY ( "Nature-based solutions"  OR  "Nature based solutions"  OR  "Nature Based Solutions"  OR  "Nature-Based Solutions"  OR  "Low impact development"  OR  "Sustainable Urban Drainage Systems"  OR  "Water Sensitive Urban Design"  OR  "Best Management Practices"  OR  "Green infrastructure"  OR  "Green blue infrastructure"  AND  "flood" )  AND  ( LIMIT-TO ( DOCTYPE ,  "ar" )  OR  LIMIT-TO ( DOCTYPE ,  "ch" )  OR  LIMIT-TO ( DOCTYPE ,  "re" )  OR  LIMIT-TO ( DOCTYPE ,  "bk" ) )  AND  ( LIMIT-TO ( LANGUAGE ,  "English" ) )".

Based on pre-search process, the number of scientific articles with respect to the concepts of NBS, LIDs, SuDS, WSUD, BMPS, GI, and BGI started increasing significantly from 2007. Therefore, the time window selected for the review process was from 1 January 2007 to 1 December 2018. 1387 articles published in scientific journals have been found in the Scopus database and the same search performed in Web of Science resulted in 1212 articles. The articles from both databased have been combined to 2599 articles. Duplicated articles found from the applied queries were then removed, resulting in a total number of 1395 articles. As consequence, the 1204 articles resulting from the search query.

**2.2 Selection process**

As stated in the Introduction, this study aims at reviewing the state of art of the research on NBS that specifically address hydro-meteorological risk reduction. In this regard, the key objectives of the present review work were carefully formulated as follows:

1) To assess the state-of-the-art in research concerning both small and large scale NBS for hydro-meteorological risk reduction;
2) To review the use of techniques, methods and tools for planning, selecting, evaluating and implementing NBS for hydro-meteorological risk reduction;
3) To review the socio-economic influence in the implementation of NBS for hydro-meteorological risk reduction as well as their multiple benefits, co-benefits, effectiveness and costs;
4) To identify trends, knowledge gaps and proposed future research prospects with respect to the above three objectives.

These key objectives defined for the review with the intention that the results could be both quantitative and qualitative.

The 1204 articles resulting from the search query (Section 2.1) were thus evaluated against their relevance with respect to these objectives, and those found of little or no pertinence with the topic removed. This selection process involved a set of progressive steps as schematized in Fig.3 and detailed below.

Initially, all articles were analysed on the basis of reading titles and keywords and their relation to the search terms. For example, articles having 'resilience', 'stormwater' or other relevant words in the title or keywords were selected for continued analysis.

Secondly, a more in-depth analysis was conducted, based on reading the abstract of each article selected in the previous step. The criteria was that the abstract should discuss about hydro-meteorological risk reduction. For example, if the abstract of the articles focuses more on water quality than risk, then that paper was excluded. This step served to reduce the number of articles from 380 to 185.

Finally, reading full articles was undertaken to identify those that were relevant to the review objectives. Any studies appearing to meet the key objectives (dealing with subjects such as effectiveness of NBS, techniques, method and tools for planning, and others which are of relevance for the key objectives) would then be included in the review. As a result, the entire selection process resulted in a total of 137 articles were selected as relevant for the objectives of the present review and were obtained for review.

**Comments from Referee 2.3** Search terms you had several search terms from your first column with "urban", this may have included a bias toward urban

**Authors' response 2.3:** We understand the Reviewer's concern, but we would like to point out that as mentioned on page 3 line 10, the concept of Nature-Based Solution was historically linke to different names in different countries [e.g. Low Impact Developments (LIDs), Best Management Practices (BMPs), Water Sensitive Urban Design (WSUD), Sustainable Urban Drainage Systems (SuDS), Green Infrastructure (GI), Blue-Green Infrastructure (BGI), Ecosystem-based Adaptation (EbA) and Ecosystem-based Disaster Risk Reduction (Eco-DRR)]. In this cases, there are only 2 search terms that include "Urban" out of the 10 search terms. Therefore, if we do not include search terms like 'Water Sensitive Urban Design' (WSUD) and 'Sustainable Urban Drainage Systems' (SuDS), we may miss some important articles related to the topic. Furthermore, only 130 of the 1387 papers from Scopus appear due to these terms and only 4 articles out of 137 were included in the review. This means that the word "urban" contributed to only 2.9% of the total 88% urban cases shown in Figure 5.a Therefore, we concluded that including these 2 terms do not have a significant impact in terms of bias. For sake of clarity, this has been now clarified also in the manuscript (section "Trends, knowledge gaps and future research prospects").

**Authors' change in the revised manuscript:**

Most of the literature to date is about NBS in urban areas whereas those contexts concerning river and coastal floods, droughts and landslides are the least addressed. 88% of all articles were concerned with runoff reduction or flood risk reduction in urban areas (Fig. 5a). It is worthwhile to notice that two out of the ten search terms in Table 1 contain the word "urban". This was in order to include two popular concepts linked to NBS for hydro-meteorological risk, which are WSUD and SuDS (cf. the overview of terminology given in Section 2). Nevertheless, the literature sourced using these two search terms only accounts for 2.5% of the total 88% urban cases shown in Figure 5.a. Therefore, no significant bias was introduced in our findings by the inclusion of the word "urban" through these two search terms.

**Comments from Referee 2.4** One of the main objectives of this review was to find trends and patterns, so only section 4.1 Trends, knowledge gaps and future research prospects provides quantitative results, the remaining sections onward are mainly qualitative descriptions to answer your pre-defined research questions: e.g. (2) Effectiveness of multiple NBS sites, etc. It should be clarified that you the review is quantitative but also qualitative based on pre-defined questions. However you do not justify why you selected these topics - again, they did not emerge as trends in the literature, you selected them and then found literature to analyse them. In other words, you combine deductive with inductive research. This should be made more explicit, or you should choose one or the other.

**Authors' response 2.4:** We thank you the reviewer for this comment which really helped us to re-shape the manuscript in a much more coherent form. As discussed earlier (comment 2.2), we have now explicitly stated that the literature material was selected to answer our pre-defined research questions. Trends, knowledge gaps and proposed future research prospects were mainly evaluated with respect to these pre-defined objectives - something that should have been evident from Table 3 but that we anyway missed to comment on in text, thus leading to confusion. For each given topic embedded in our key research questions, this Table specifies the number of articles found that deal with it and it summarizes the knowledge gaps and future research prospects drawn upon them. Trends and path - as emerging from those articles – are therefore discussed not in general, but with respect to each of these topics, which was the criterion based on which Section 4 was divided into subsections. The different sub-sections are meant to reflect the key objectives defined for the review with the intention that the results could be both quantitative and qualitative

In the revised manuscripts, we will also slightly modify the titles and contents of some subsections of Section 4 to better highlight the correspondence between them and the research questions of this review. Furthermore, we will move Section 4.1 "Trends, knowledge gaps and future research prospects" to end of Section 4, as we feel this will better clarify the logic of the paper. Here we also plan to include a paragraph to explicitly comment on Table 3 and to better highlight the quantitative results emerging from our analysis. Finally, we will expand the "Introduction" Section to better motivate our research questions' choice.

**3. Other**

**Comments from Referees 3.1** Some paragraphs appeared to be more a promotion of author's projects rather than related to the literature review ?? They might belong in the conclusions but not as part of the analysis.

**Authors' response 3.1** We apologize if some paragraphs appeared to be more a promotion of author's projects. Paragraph on page 10, line 12 has been relocated to conclusion.

**Comments from Referees 3.2** The manuscript needs to be redrafted by a native English speaker. e.g. p8, line 27 "desiderative" ;)

**Authors' response 3.2** Thank you for suggestion. The revised manuscript has been reviewed by a native English speaker.

**Comments from Referees 3.3** The table on websites related to the topic is good but excludes a few important sites, namely IUCN's data base on EbA projects and the Partnership for Environment and Disaster Risk Reduction (PEDRR) website.

**Authors' response 3.3** We apologize for the missing site lists. IUCN's database on EbA projects and the Partnership for Environment and Disaster Risk Reduction (PEDRR) website have been included in Table 4.

---

## Author Comment (AC2) · 20 Aug 2019

**Reponses to second referee's comments on "Nature-Based Solutions for hydro-meteorological risk reduction: A state-of-the-art review of the research area" by Laddaporn Ruangpan et al.**

**Comments from Referee:** Summary of the manuscript This manuscript (ms) reviews scientific publication on Nature-based solutions (NBS) for hydro-meteorological risk reduction and related terms. The authors proceeded in a systematic way by using search terms in various scientific literature databases and analyzed over 1000 references. The ms concludes by summarizing the main findings and suggesting further research in some of the reviewed areas. Evaluation I think the topic of this manuscript is highly relevant and important in order to review NBS to tackle the ecological crisis the world is facing. Accordingly, I do think that this ms should be considered for publication. However, I have major doubts if the presented ms really helps to summarize the vast amount of literature on NBS and if it really identifies the knowledge gap in order to be able to recommend the area of focus for future research. My main concerns are the following:

**Authors' response:** Thank you for your encouragement and comments. Your concerns are addressed in this response letter. Please find our point-by point response below.

**i) Methodology**

**Comments from Referee:** a simple search for "Nature-based solutions" in the WoS shows that three of the four most relevant and most cited papers have not been considered in this ms (Keesstra et al. 2018, Nesshover et al. 2017, and Eggermont et al. 2015). Accordingly, I would recommend revising the method of selecting research articles that are being taken into account in the review.

**Authors' response**: Thank you for your suggestion to make this review more complete. Yes indeed, a simple search for "Nature-based solutions" in the WoS shows that these three papers that the review is referring to did come up in the search and they are indeed among the most cited ones and -without doubt- of relevance for the general subject of NBS. However,we would like to clarify that the goal of our study is to not review the state of the art on all NBS terms (i.e. SuDs, WSUD, BMP, GI etc.) in general, rather to specifically investigate how Nature-based solutions have been used or studied to reduce hydro-meteorological risk. Therefore, the search terms had to simultaneously include on terms for "Nature-based solutions" and one term for hydro-meteorological risk as risk was one of the critieria used to filter the total number of articles (over 6,300). For that reason, Nesshöver et al., (2017) and Eggermont et al., (2015) are not shown in this case and were not taken forward for a more detailed analysis in the 'Finding' section. On the other hand, having recognized the relevance of those articles with respect to the general topic of NBS, they will be included in "Overview of definitions and theoretical backgrounds" in the revised version. This section is not part of the Findings section. On the contrary, Keesstra et al., (2018) has now been included as it fulfils the search criteria mentioned above. Note that following Reviewer 1's suggestion, we have expanded the literature search beyond Scopus, by including Web of Science database. This has made more articles, including Keesstra et al., (2018)", discoverable.

**ii) Structure**

**Comments from Referee:** I recommend limiting the structure to three levels of subsection: especially section 4 could be better structured, avoiding sections with titles that do not clearly adhere to a three-level subsection structure.

**Authors' response:** Thank you very much for pointing out the structural issues. The authors have limited the structure to three levels of the subsection. The different sub-sections in Section 4 are meant to reflect the 4 key objectives defined for the review with the intention that the results could be both quantitative and qualitative.

**iii) Content**

**Comments from Referee:** Content is more valuable than academic metrics: while I do see a value in using academic metrics and search engines to select relevant literature, it would be helpful to review the actual characteristics, benefits, and scales of various NBS. Specifically, it would be helpful to have a table that summarizes area, volume of water retention, costs, and effectiveness (and other characteristics) of different NBS. The number of articles does not indicate anything about the effectiveness of a NBS, accordingly, I would encourage the authors to focus more on the characteristics of NBS rather than the number of articles found. In short, more quantitative assessments of the benefits of NBS rather than generic statements would be highly appreciated.

**Authors' response:** In the revised version, we have investigated further those aspects and carried out a a more quantitative assessment of NBS for hydro-meteorological risk reduction. A new table has been included, which summarizes effectiveness, benefits and costs of different NBS based on the case studies found in the reviewed literature. The table is given below.
We agree that such quantitative information are indeed very valuable and thank the Reviewer for this input. Neverthless, we also see value in using academic metrics: although the number of articles does not indicate anything about the effectiveness of NBS, it provides indications on the direction and the degree of advancement of the research done on this specific topic, which is one of the review objectives.

**Authors' change in the revised manuscript:** A summary of effectiveness, co-benefits and cost of NBS measures at small scale is shown in Table 4 and at large scale is shown in Table 5

**Table 4: Summary of effectiveness, co-benefits and costs of small scale NBS measures**

| Measures | References | Case studies | Area/ volume covered by NBS | Effectiveness | | Co-benefits | Cost/ m²* | Remark |
|---|---|---|---|---|---|---|---|---|
| | | | | Runoff volume reduction | Peak flow reduction | | | |
| **Porous pavement** | Shafique et al., (2018) | Seoul, Korea | 1050 m² | ~30–65% | - | • Removing diffuse pollution • Enhancing recharge to groundwater | ~$252 | ‣ More effective in heavier and shorter rainfall events. |
| | Damodaram et al., 2010 | Texas, USA | 2.99 km² | - | ~10% - 30% | | | |
| **Green roofs** | (Burszta-Adamiak and Mrowiec, 2013) | Wroclaw, Poland | 2.88 m² | - | 54%-96% | • Reducing nutrient loadings. • Saving energy • Reducing air pollution • Increasing amenity value | ~$564 | ‣ More efficient in smaller storm events than larger storm events |
| | (Ercolani et al., 2018) | Milan, Italy | 0.39 km² | ~15%-70% | ~10-80% | | | |
| | (Carpenter and | Michigan, USA | 325. 2 m² | ~68.25% | ~88.86% | | | |

| Measures | References | Case studies | Area/volume covered by NBS | Effectiveness | | Co-benefits | Cost/m²* | Remark |
|---|---|---|---|---|---|---|---|---|
| | | | | Runoff volume reduction | Peak flow reduction | | | |
| | Kaluvakola nu, 2011) | | | | | | | |
| **Rain gardens** | (Ishimatsu et al., 2017) | Japan | 1.862 m² | ~36-100% | - | • Providing a scenic amenity. | ~$501 | ‣ More effective in dealing with small discharges or rainwater |
| | (Goncalves et al., 2018) | Joinville, Brazil | 34,139 m² | 50% | 48.5% | • Increasing the median property value • Increasing biodiversity | | |
| **Vegetated swales** | (Luan et al., 2017) | Beijing, China | 157 m³ | ~0.3–3.0%. | 2.2% | • Reducing concentrations of pollutants • Increasing biodiversity | ~$371 | ‣ More effective in heavier and shorter rainfall events. |
| | (Huang et al., 2014) | Haihe River basin, China | 1,500 m³ | 9.60% | 23.56% | | | ‣ Not suitable in mountains areas |
| **Rainwater harvesting** | (Khastagir and Jayasuriya, 2010) | Melbourne, Australia | 1 m³ -5 m³ | ~57.8%-78.7% | - | • Improving water quality (TN was reduced around 72%-80%) | ~$865 /m³ | |
| | (Damodara m et al., 2010) | Texas, USA | 1.5 km² | - | ~8%-10% | | | |
| **Dry detention pond** | (Liew et al., 2012) | Selangor, Malaysia | 65,000 m² | - | 33-46% | • Providing recreational benefits. | | ‣ Delaying the time to peak by 40-45 min |
| **Detention pond** | (Damodara m et al., 2010) | Texas, USA | 73,372 m³ | - | ~20% | • Providing biodiversity benefits | ~$60 | |
| | (Goncalves et al., 2018) | Joinville, Brazil | 9,700 m³ | 55.7% | 43.3% | • Providing recreational benefits. | | |
| **Bio-retention** | (Luan et al., 2017) | Beijing, China | 945.93 m³ | ~10.2–12.1%. | - | • Reducing TSS pollution • Reducing TP pollution | ~$534 | • Measure has a better reduction effectiveness in various rainfall intensities. |
| | (Huang et al., 2014) | Haihe River basin, China | 1,708.6 m³ | 9.10% | 41.65% | | | |
| | Khan et al., 2013; | Calgary | 48 m³ | ~90% | - | | | |
| **Infiltration trench** | (Huang et al., 2014) | Haihe River, China | 3,576 m³ | 30.80% | 19.44% | • Reducing water pollutant • Improving surface water quality. | ~$74 | |
| | (Goncalves et al., 2018) | Joinville, Brazil | 34,139 m² | 55.9% | 53.4% | | | |
| **Street trees** | (Soares et al., 2011) | Lisbon, Portugal | 41,247 street trees | | | • Net benefit €6.55 million per annual of benefits | €45.6 per annual | |
| **Green roof and Porous pavement** | (Damodara m et al., 2010) | Texas, USA | 4.49 km² | - | ~10%-35% | • Saving energy • Increasing amenity value | | • More effective in smaller events |
| **Swale and Porous pavement** | (Behroozi et al., 2018) | Tehran, Iran | - | 5%-32% | ~10%-21% | ‣ Decreasing TSS pollution 50-60% | | More effective in smaller events |
| **Rainwater harvesting and Porous pavement** | (Damodara m et al., 2010) | Texas, USA | 4.49 km² | - | 20%-40% | • Removing diffuse pollution | | • More effective in smaller events |
| **Detention pond and Raingarden** | (Goncalves et al., 2018) | Joinville, Brazil | 18,327 m² | 70.8% | 60.0% | • Providing a scenic amenity. | | |

| Measures | References | Case studies | Area/ volume covered by NBS | Effectiveness | | Co-benefits | Cost/ m²* | Remark |
|---|---|---|---|---|---|---|---|---|
| | | | | Runoff volume reduction | Peak flow reduction | | | |
| **Detention pond and Infiltration trench** | (Goncalves et al., 2018) | Joinville, Brazil | 18,327 m² | 75.1% | 67.8% | Improving surface water quality. | | |

**\*Remark** Cost of each measure is based on (CNT, 2009; Nordman et al., 2018; De Risi et al., 2018)

**Table 5: Summary of effectiveness, co-benefits and costs of large scale NBS measures**

| Measures | References | Case studies | Area/ volume covered by NBS | Effectiveness | Co-benefits | Cost |
|---|---|---|---|---|---|---|
| **De-culverting (river restoration)** | (Chou, 2016) | Laojie River, Taiwan | 3 km | • It can reduce flood risk up to 100 year return period | • Increasing landscape value
 • Increasing recreational value | ~$18.6 million |
| **Floodplain lowering** | (Klijn et al., 2013). | Deventer Netherlands | 5.01 km² | • It can reduce water level 19 cm | • Increasing nature area
 • Increasing agriculture value | ~€136.7 million e |
| **Dike relocation/ floodplain lowering** | (Klijn et al., 2013). | Nijmegen/ Lent, Netherlands | 2.42 km² | • It can reduce water level 34 cm | • Increasing floodplain area
 • Increasing recreational value | ~€342.60 million |
| **Floodwater storage** | (Klijn et al., 2013). | Volkenrak-Zoommeer | 200 million m³ | • It can reduce water level 50 cm | • Increasing habitat and biodiversity in the area
 • Increasing recreational value | ~€386.20 million |
| **Green floodway** | (Klijn et al., 2013). | Veessen-Wapenveld | 14.10 km² | • It can reduce water level 71 cm | • Increasing floodplain area
 • Increasing recreational value | |
| **Wetlands (Mangroves and salt Marshes)** | (Coppenolle, 2018; Gedan et al., 2011) | | | • It can mitigate storm surge 80%
 • It can protect against tsunami impacts | • Providing shoreline protection services | |
| **Forest rapirian buffer, basins and ponds and coarse woody debris** | (McVittie et al., 2018) | Pickering, North Yorkshire, UK | 68,6 km² | • Increased water storage 90,000-138,000 m³
 • Peak flow rate reduction 6.7–14.7%
 • Flood peak delay by 20 min | • Increase habitat creation value
 • Education and knowledge
 • Community development
 • A benefit/cost ratio of 4.98 | ~€1.58 million |
| **Renaturation** | (McVittie et al., 2018; NWRM, 2019) | Seymaz river, Switzerland | 0.4 km² | • Water storage 800,000 m³ | • | ~€61 million (€76.3/m³) |

**iv) Definitions**

**Comments from Referee:** in my opinion, it would be helpful to provide a table with definitions and examples of the various academic terms used in the review: The study provides generic definitions for GI, EbA, and NBS, but it is left upon the reader to interpret the definitions. I would recommend to complement Table 2 with some quantitative figures on water retention, area, costs, advantages, disadvantages etc.... (see also the previous comment).

**Authors' response:** Thank you for the comment and suggestion. Referee 1 also raised similar comments on the definitions. We have included more explanation on the definition in section 3 (now section 2) "Overview of definitions and theoretical backgrounds on the terminology of NBS" Here we also recommend the reader to refer to Nesshöver et al., (2017). and others works for a more exhaustive analysis on terminology, which is beyond the goal of our study. We feel

that this additions, together with the newly provided Tables 4 and 5, should provide the reader enough guidance in the interpretation of our results.

**Authors' change in the revised manuscript** (revised and added texts have yellow highlights)**:**

[revised manuscript text omitted]

different interpretations. It can be useful as they may be easier to encourage stakeholders to take part in the discussion.

Moreover, features of NBS provide an alternative to work with existing measures or grey infrastructures. Therefore, it is important to note that very often a combination between natural and traditional engineering solutions (a.k.a. "hybrid" solutions) is likely to produce more effective results than any of these measures alone, especially when their co-benefits are taken into consideration (Alves et al., 2019).

Important advances in the science and practice of NBS is provided by the EKLIPSE Expert Working Group, who developed the first version of a multiple-dimension impact evaluation framework to support planning and evaluation of NBS projects. The document includes a list of impacts, indicators and methods for assessing the performance of NBS specifically at the urban scale (EKLIPSE, 2017). Lafortezza et al., (2018) has also reviewed different case studies around the world where NBS have been applied from micro-scale to macro-scale. Furthermore, an overview on how different types of nature based solutions can regulate to ecosystem services (i.e., soil protection, water quality, flood regulation, and water provision) has been carried out by Keesstra et al., (2018).

**v) Drought**

**Comments from Referee** it is well know that land reclamation and restoration reduces evaporation and mitigates the drought risk. However, the authors found only one single study referring to the drought risk. This might be due to a methodology based on "key words" rather than content.

**Authors' response:**

Thank you very much for your suggestion. While the methodology helps limit the scope of the paper, the authors understand that it may also cause some gaps in the study. However, the authors have attempted to review papers as comprehensively as possible to mitigate this issue.

Since the aim of the study is to review Nature Based Solutions and those terms used in conjunction with NBS, additional terminology like reclamation and restoration were not specifically used. This search term could introduce a bias, as the authors then assume the solutions before the review. To be transparent, we have included a sentence to acknowledge that the selection of key search words can limit the hydro-meterological risk measures that appear in returned papers.

Nevertheless, the authors believe that the method still provides useful direction for a state of the art review and defining research gaps on Nature-Based Solutions for Hydro-meteorological risk reduction.

**vi) Scale and examples**

**Comments from Referee** one example that struck me is the NBS "Room for the River Programme" in the Netherlands at the Rhine and Meuse. It is general knowledge that flood protection has to start upstream in the headwaters, where most of the precipitation occurs, to be efficient. Nevertheless, the ms only mentions NBS in the Netherlands (a third of the Netherlands are below sea level and sea levels are rising), ignoring the far more relevant NBS in upstream

countries. This might be linked to the somewhat limited methodology of the literature review (see comment i).

**Authors' response**: Thank you very much for your comment. The requirements of this articles that is to focus on peer-reviewed articles in English and we agree with the reviewer that upstream cases exist, which the authors had conducted additional research based on the comment of the reviewer.

It can be summarized that the EU Flood directive specifies that countries upstream or downstream should avoid taking measures that will increase the flood risk to other countries in the same river catchment. In case this is not feasible, the countries should consult with the other member states to agree to the proposed measures (EU, 2019). As far as the authors are aware, there is a project in the Rhine basin called Adaptive Land use for Flood Alleviation (ALFA). "Room for the River Programme" by the Ducth is also part of this project. However, all of the project's documents from upstream in Germany that the authors have found are only in the grey literatures and in the German language, which are out of the scope of this article.

On the other hand, there are many documents and publications on the "Room for the River Programme" that are available in English. Moreover, "Room for the river programme" is one of the big projects on a large scale NBS which has been successfully studied and implemented it can be used as an example to other countries.

**vii) Tools**

**Comments from Referee:** in my opinion, the review of tools could be shortened, as it is slightly off the topic. Instead, more attention could be given to the quantification of the various benefits of NBS could be provided (see comment iii).

**Authors' response:** Authors have shortened the review of tools. However, the leading message should still be included since the tools are important for selection, evaluation and operation of NBS. One of the purposes of this review is to review the use of techniques, methods and tools for planning, selecting, evaluating and implementing NBS. The benefit of this section is to provide information to the reader as to what the available tools are that can be used for a specific purpose.

The authors have also included some quantification of NBS measures as suggested by the reviewer in Table 4 and 5 as shown in comment iii and benefits of NBS also discuss in section 4.5 in manuscripts. However, discussing the quantitative co-benefits of NBS is still very challenging as there is a lack of information on assessment quantitative value of the ecosystem. Such challenges and limitations will be explictely commented on in the revised version.

**viii) Conclusion**

**Comments from Referee:** the current conclusion provides general and generic statements and any reader somewhat familiar with the topic does not really learn anything new. It would be helpful to generate more conclusive and quantitative statement based on the review: which NBS are most effective, which provide most multi-benefits, which require least areas, which are most accepted?

**Authors' response:** Thank you for the suggestion. The authors have revised the conclusion to summarize the quantitative statement of NBS on; "which NBS are most effective, which

provide most multi-benefits, which require the least area, which are the most accepted" as the authors suggested.

However, this has proven to be very difficult because the effectiveness, benefits and acceptance of NBS are dependent on the implementation purposeslocal context and cultural setting. For example, small NBS are more suitable for urban flooding while large scale NBS are more suitable for river floods, coastal floods, droughts and landslides. Large scale NBS can provide more benefits compared to small scale NBS because it has a bigger space, thus more function can be included in the design process. For example, Laojie river project in Taoyuan City in Taiwan changed the channel into an accessible green corridor. This project helps in reducing flood risk, improving riverside landscapes, increasing recreation area, increasing the aesthetic value in the area, and improving river water quality. On the other hand, small scale NBS need less area because most of the measures can be implemented in the free space. For example, green roofs can be implemented on the roofs of buildings, and permeable pavements can be implemented in car parks. Investments in NBS will benefit society by providing cost-effective measures and adaptive strategies that protect their communities and achieve a range of co-benefits. Therefore, bridging the gaps between researchers, engineers and stakeholders will help to improve the capacity of NBS in reducing hydro-meteorological risk as well as increasing thier benefits. Strengthening these aspects may be beneficial for improving acceptance of NBS at the local level.

In the revised version, all of the above information has been included in the conclusion section, and a summary of quantitative information on effectiveness, co-benefits and cost for different NBS measures can be found in Table 4 and 5 in the revision.

---

## Author Response (AR1)

**Reponses to editor's comments on "Nature-Based Solutions for hydro-meteorological risk reduction: A state-of-the-art review of the research area" by Laddaporn Ruangpan et al.**

As you know, two reviewers have now provided detailed reviews, which you have replied in detail to. Both reviewers recommended major revisions and highlighted serval major issues regarding the design of your research, the chosen methods and evaluation criteria as well as the added-value for the scientific community.

In particular, in addition to mentioned issues by the reviewers, I kindly ask you to include in your manuscript more explicit also the terms of hazards and risks because the different communities addressing NBS also define differently the following terms related their research field

**Authors' response:** Thank you for your encouragement and comments. Your concerns are addressed in this response letter and the manuscript revised accordingly. Please find our point-by-point response below.

**1. Comments from the editor:** hydro-meteorological hazard vs hydro-meteorological risk vs hydro-meteorological disaster: It seems throughout the manuscript that you use hazard and risk synonymously (e.g. p5 l10). Furthermore, which hazards types did you include/select in your review considering the brought spectrum of hydro-meteorological hazards (e.g. storms, hail, snow avalanches, flash floods, …). Please provide here further explanations.

**Authors' response:** The authors have explained both terms hydro-meteorological hazard and hydro-meteorological risk more explicitly in abstract and introduction (page 1, line 13-15 and line 27-31). The hazards that have been included in this review are floods, droughts, storm surges, and landslides (page 2, line 24-25 and page 5, line 6).

**2. Comments from the editor:** vulnerability

**Authors' response:** Key references are now provided further explanation of vulnerability since it is not the main part of this work and only briefly mentioned in relation to hydro-meteorolgical risk.

3. **Comments from the editor:** climate change adaptation and disaster risk reduction

**Authors' response:** In the revised version**,** climate change adaptation and disaster risk reduction have been explained on page 2, line 2-4.

3. **Comments from the editor:** Moreover, I wonder why you did not include the term 'risk' in your second order concept for the literature research even 'risk reduction' is mentioned in the title of your manuscript.

**Authors' response:** Thank you very much for your comment. In the revised version, we have revised the methods, thus now the term 'risk' have been included.

**Reponses to first referee's comments on "Nature-Based Solutions for hydro-meteorological risk reduction: A state-of-the-art review of the research area" by Laddaporn Ruangpan et al.**

The premise of this article is extremely interesting and some of the conclusions of the article, in particular the "Overview of knowledge gaps / potential future research" is a very useful contribution to advancing this topic. The article helps to give an overview of the many concepts and terms associated with Nature based solutions for disaster risk reduction and it attempts to provide a mixed quantitative /qualitative assessment of a number of pre-determined questions that the authors have outlined as the objectives of the review. It therefore merits to be published if some fundamental methodological issues can be resolved.

**Authors' response:** Thank you for your encouragement and comments. Your concerns are addressed in this response letter and the manuscript revised accordingly. Please find our point-by point response below.

**1. Concepts**

**Comments from Referee:** The article provides an interesting historical overview of the different related concepts but there is still a confusion of terms. The abstract in particular is confusing, i.e. Nature based Solutions (NbS) is generally considered to be an umbrella term under which other types of approaches, Eba, Eco-DRR and GI / GBI provide more specific solutions to more specific issues (see various definitions given by IUCN and EU-related). This does not come out clearly in the article.

For example: p. 4/ line 30 NbS is not just about storm water

**Authors' response:** Thank you for pointing out this issue. We agree that terminology was confusing in the Abstract and other instances. This has been clarified in the revised manuscript. Furthermore, Section 3 "Overview of definitions and theoretical backgrounds", has been modified and expanded to better highlight the definition of NBS as an umbrella concept, as the reviewer suggested. This section also has been relocated to section 2 before "Materials and methodology" section as it discusses more on the background of NBS.

P.4 line 30: revised. Now, we specifically refer to SuDs, LIDs and WSUD terms in the sentence.

**Authors' change in the revised manuscript:**

[revised manuscript text omitted]

**2. Methodology of the review**; This is where this reviewer has the greatest number of questions:

**Comments from Referees 2.1** Good that multiple data bases were used but why assume that just because Scopus has the greatest number of articles, that it is the most comprehensive? You could have merged all three searches and then removed duplicates.

**Authors' response 2.1** Thank you very much for your comment. The authors have revised the methodology (see also next comment) by including both Web of Science and Scopus databases and merged the two searches together as recommended by the reviewer, and removed duplicates. Note that Google Scholar has been completely excluded from the revised methodology because it has limited metadata and filters which, at present, do not allow to limit results to articles published in peer-reviewed, scientific journals written in English (one of the three selection criteria adopted in our search process).

**Comments from Referees 2.2** Adding missing articles adds a huge bias to your search. Which articles were selected and based on what criteria? That the keywords were there? - Which criteria were used for deleted certain articles - perhaps I missed this?

**Authors' response 2.2** We agree that the methodology of this review was not clearly explained and had some flaws. Thanks to Reviewer's comments, our methodological approach has been carefully revised and improved. Specifically:

1) Bias introduced by missing articles has been removed, namely those articles are no longer evaluated neither included/added in the analysis. Note that few comments drawn upon this subset of articles have been retained because considered of relevance to our discussion, but they are now included in the new Section 2, which is not part of the "Findings" section.

2) An analysis of why other papers in the extended list did not appear in the search shows that they were missed because they use the terms 'green and grey infrastructure' as opposed to 'green infrastructure'

directly. As this is merely a language issue, the term 'green and grey infrastructure' was added to the search terms.

3) As this Reviewer pointed out, the selection process was not clearly explained in the original manuscript. We have now substantially expanded the methodological section, by explicitly stating the objectives of the review and by explaining the criteria used for selecting the literature of relevance with respect to these objectives. This is summarized in the diagram below (included in the new version of the manuscript) which shows that the method consists of two phases. For the search process (phase I) the only selection criteria adopted were that (a) articles are published in peer-reviewed and scientific journals written in English; (b) articles reported on NBS in terms of hydro-meteorological risk reduction (construction of the search query based on the keywords in Table 1); (c) articles were published in the period 2007 to 1 December 2018. The search process resulted in a total of 1204 articles which were then subjected to selection process (Phase II). The selection process involved a set of progressive steps as schematized in Fig.3 and detailed in the following: << *Initially, all articles were analysed on the basis of reading titles and keywords and their relation to the search terms. For example, if titles and keywords of articles were not considered relevant because of their complete titles, or because the keywords did not match closely enough to the topics, they were omitted. This step served to reduce the number of articles from 1395 to 30. Secondly, a more in-depth analysis was conducted, based on reading the abstract of each article selected in the previous step. The criteria at this step was that the abstract should discuss hydro-meteorological risk reduction. For example, if the abstract of the articles focused more on water quality than risk, then that paper was excluded. This step served to reduce the number of articles from 380 to 185. Finally, articles were read in full to identify those that were relevant to the review objectives. Any studies appearing to meet the key objectives (dealing with subjects such as effectiveness of NBS, techniques, method and tools for planning, and others subjects relevant to the key objectives) would then be included in the review. As a result, the entire selection process resulted in a total of 137 articles relevant to the objectives of the present review.*>> (text extrapolated from the revised Section 2.2 (now Section 3.2)). For the sake of completeness and clarity, the new version of the entire methodological section is provided below.

**Authors' change in the revised manuscript:**

**3. Materials and methodology**

[revised manuscript text omitted]

**Comments from Referee 2.3** Search terms you had several search terms from your first column with "urban", this may have included a bias toward urban

**Authors' response 2.3:** We understand the Reviewer's concern, but we would like to point out that as mentioned on page 3 line 10, the concept of Nature-Based Solution was historically linke to different names in different countries [e.g. Low Impact Developments (LIDs), Best Management Practices (BMPs), Water Sensitive Urban Design (WSUD), Sustainable Urban Drainage Systems (SuDS), Green Infrastructure (GI), Blue-Green Infrastructure (BGI), Ecosystem-based Adaptation (EbA) and Ecosystem-based Disaster Risk Reduction (Eco-DRR)]. In this cases, there are only 2 search terms that include "Urban" out of the 10 search terms. Therefore, if we do not include search terms like 'Water Sensitive Urban Design' (WSUD) and 'Sustainable Urban Drainage Systems' (SuDS), we may miss some important articles related to the topic. Furthermore, only 130 of the 1387 papers from Scopus appear due to these terms and only 4 articles out of 137 were included in the review. This means that the word "urban" contributed to only 2.9% of the total 88% urban cases shown in Figure 5.a Therefore, we concluded that including these 2 terms does not have a significant impact in terms of bias. For sake of clarity, this has been now clarified also in the manuscript (section "Trends, knowledge gaps and future research prospects").

**Authors' change in the revised manuscript:**

The review of the 137 articles indicates that most of the research to date has been carried out in an urban context, whereas the contexts concerning river and coastal floods, droughts and landslides are the least addressed. More specifically, 88% of all articles deal with runoff reduction or flood risk reduction in urban areas (Fig. 5b). It is worthwhile to notice that two out of the ten search terms in Table 2 contain the word "urban". This was in order to include two popular concepts linked to NBS for hydro-meteorological risk, which are WSUD and SUDs (cf. the overview of terminology given in Section 2). Nevertheless, the literature sourced using these two search terms only accounts for 2.9% of the total 88% urban cases shown in Figure 5b. Therefore, no significant bias was introduced in our findings by the inclusion of the word "urban" through these two search terms.

**Comments from Referee 2.4** One of the main objectives of this review was to find trends and patterns, so only section 4.1 Trends, knowledge gaps and future research prospects provides quantitative results, the remaining sections onward are mainly qualitative descriptions to answer your pre-defined research questions: e.g. (2) Effectiveness of multiple NBS sites, etc. It should be clarified that you the review is quantitative but also qualitative based on pre-defined questions. However you do not justify why you selected these topics - again, they did not emerge as trends in the literature, you selected them and then found literature to analyse them. In other words, you

combine deductive with inductive research. This should be made more explicit, or you should choose one or the other.

**Authors' response 2.4:** We thank you the reviewer for this comment which really helped us to re-shape the manuscript in a much more coherent form. As discussed earlier (comment 2.2), we have now explicitly stated that the literature material was selected to answer our pre-defined research questions. Trends, knowledge gaps and proposed future research prospects were mainly evaluated with respect to these pre-defined objectives - something that should have been evident from Table 3 but that we had negelcted to comment on in text, thus leading to confusion. For each given topic embedded in our key research questions, this Table specifies the number of articles found that deal with it and it summarizes the knowledge gaps and future research prospects drawn upon them. Trends and path - as emerging from those articles – are therefore discussed not in general, but with respect to each of these topics, which was the criterion based on which Section 4 was divided into subsections. The different sub-sections are meant to reflect the key objectives defined for the review with the intention that the results could be both quantitative and qualitative

In the revised manuscripts, we also slightly modify the titles and contents of some subsections of Section 4 to better highlight the correspondence between them and the research questions of this review. Furthermore, we have moved Section 4.1 "Trends, knowledge gaps and future research prospects" to end of Section 4, as we feel this is better clarify the logic of the paper. Here we also plan to include a paragraph to explicitly comment on Table 3 and to better highlight the quantitative results emerging from our analysis. Finally, we have expanded the "Introduction" Section to better motivate our research questions' choice.

**3. Other**

**Comments from Referees 3.1** Some paragraphs appeared to be more a promotion of author's projects rather than related to the literature review ?? They might belong in the conclusions but not as part of the analysis.

**Authors' response 3.1** We apologize if some paragraphs appeared to be more a promotion of author's projects. Paragraph on page 10, line 12 has been relocated to conclusion.

**Comments from Referees 3.2** The manuscript needs to be redrafted by a native English speaker. e.g. p8, line 27 "desiderative" ;)

**Authors' response 3.2** Thank you for suggestion. The revised manuscript has been reviewed by a native English speaker.

**Comments from Referees 3.3** The table on websites related to the topic is good but excludes a few important sites, namely IUCN's data base on EbA projects and the Partnership for Environment and Disaster Risk Reduction (PEDRR) website.

**Authors' response 3.3** We apologize for the missing site lists. IUCN's database on EbA projects and the Partnership for Environment and Disaster Risk Reduction (PEDRR) website have been included in Table 4.

**Reponses to second referee's comments on "Nature-Based Solutions for hydro-meteorological risk reduction: A state-of-the-art review of the research area" by Laddaporn Ruangpan et al.**

**Comments from Referee:** Summary of the manuscript This manuscript (ms) reviews scientific publication on Nature-based solutions (NBS) for hydro-meteorological risk reduction and related terms. The authors proceeded in a systematic way by using search terms in various scientific literature databases and analyzed over 1000 references. The ms concludes by summarizing the main findings and suggesting further research in some of the reviewed areas. Evaluation I think the topic of this manuscript is highly relevant and important in order to review NBS to tackle the ecological crisis the world is facing. Accordingly, I do think that this ms should be considered for publication. However, I have major doubts if the presented ms really helps to summarize the vast amount of literature on NBS and if it really identifies the knowledge gap in order to be able to recommend the area of focus for future research. My main concerns are the following:

**Authors' response:** Thank you for your encouragement and comments. Your concerns are addressed in this response letter. Please find our point-by point response below.

**i) Methodology**

**Comments from Referee:** a simple search for "Nature-based solutions" in the WoS shows that three of the four most relevant and most cited papers have not been considered in this ms (Keesstra et al. 2018, Nesshover et al. 2017, and Eggermont et al. 2015). Accordingly, I would recommend revising the method of selecting research articles that are being taken into account in the review.

**Authors' response:** Thank you for your suggestion to make this review more complete. Yes indeed, a simple search for "Nature-based solutions" in the WoS shows that these three papers that the review is referring to did come up in the search and they are indeed among the most cited ones and -without doubt- of relevance for the general subject of NBS. However,we would like to clarify that the goal of our study is to not review the state of the art on all NBS terms (i.e. SuDs, WSUD, BMP, GI etc.) in general, rather to specifically investigate how Nature-based solutions have been used or studied to reduce hydro-meteorological risk. Therefore, the search terms were required to simultaneously include one term for "Nature-based solutions" and one term for hydro-meteorological risk as risk was one of the critieria used to filter the total number of articles (over 6,300). For that reason, Nesshöver et al., (2017) and Eggermont et al., (2015) are not shown in this case and were not taken forward for a more detailed analysis in the 'Finding' section. On the other hand, having recognized the relevance of those articles with respect to the general topic of NBS, and therefore are included in "Overview of definitions and theoretical backgrounds" in the revised version. This section is not part of the Findings section. On the contrary, Keesstra et al., (2018) has now been included as it fulfils the search criteria mentioned above. Note that following Reviewer 1's suggestion, we have expanded the literature search beyond Scopus, by including Web of Science database. This has made more articles, including Keesstra et al., (2018)", discoverable.

**ii) Structure**

**Comments from Referee:** I recommend limiting the structure to three levels of subsection: especially section 4 could be better structured, avoiding sections with titles that do not clearly adhere to a three-level subsection structure.

**Authors' response:** Thank you very much for pointing out the structural issues. The authors have limited the structure to three levels of the subsection. The different sub-sections in Section 4 are meant to reflect the 5 key objectives defined for the review with the intention that the results could be both quantitative and qualitative.

**Author's changes in manuscript:** We also changed the heading of section 4.2 from "small and large scale NBS for hydro-meteorological risk reduction" to "Lessons from research on small and large scale NBS for hydro-meteorological risk reduction", section 4.2.1 from "Small scale NBS" to "Research on Small scale NBS for hydro-meteorological risk reduction" and section 4.2.2 from "Large scale NBS" to "Research on Large-scale NBS for hydro-meteorological risk reduction."

**iii) Content**

**Comments from Referee:** Content is more valuable than academic metrics: while I do see a value in using academic metrics and search engines to select relevant literature, it would be helpful to review the actual characteristics, benefits, and scales of various NBS. Specifically, it would be helpful to have a table that summarizes area, volume of water retention, costs, and effectiveness (and other characteristics) of different NBS. The number of articles does not indicate anything about the effectiveness of a NBS, accordingly, I would encourage the authors to focus more on the characteristics of NBS rather than the number of articles found. In short, more quantitative assessments of the benefits of NBS rather than generic statements would be highly appreciated.

**Authors' response:** In the revised version, we have investigated further those aspects and carried out a a more quantitative assessment of NBS for hydro-meteorological risk reduction. A new table has been included, which summarizes effectiveness, benefits and costs of different NBS based on the case studies found in the reviewed literature. The table is given below.

We agree that such quantitative information are indeed very valuable and thank the Reviewer for this input. Nevertheless, we also see value in using academic metrics: although the number of articles does not indicate anything about the effectiveness of NBS, it provides indications on the direction and the degree of advancement of the research done on this specific topic, which is one of the review objectives.

**Author's changes in manuscript:** A summary of effectiveness, co-benefits and cost of NBS measures at small scale is shown in Table 4 and at large scale is shown in Table 5

**Table 4: Summary of effectiveness, co-benefits and costs of small scale NBS measures**

[revised manuscript text omitted]

| Measures | References | Case studies | Area/ volume covered by NBS | Effectiveness | Co-benefits | Cost |
|---|---|---|---|---|---|---|
| **De-culverting (river restoration)** | (Chou, 2016) | Laojie River, Taiwan | 3 km | • It can reduce flood risk up to 100 year return period | • Increasing landscape value
• Increasing recreational value | ~$18.6 million |
| **Floodplain lowering** | (Klijn et al., 2013). | Deventer Netherlands | 5.01 km² | • It can reduce water level 19 cm | • Increasing nature area
• Increasing agriculture value | ~€136.7 million e |
| **Dike relocation/ floodplain lowering** | (Klijn et al., 2013). | Nijmegen/ Lent, Netherlands | 2.42 km² | • It can reduce water level 34 cm | • Increasing floodplain area
• Increasing recreational value | ~€342.60 million |
| **Floodwater storage** | (Klijn et al., 2013). | Volkenrak-Zoommeer | 200 million m³ | • It can reduce water level 50 cm | • Increasing habitat and biodiversity in the area
• Increasing recreational value | ~€386.20 million |
| **Green floodway** | (Klijn et al., 2013). | Veessen-Wapenveld | 14.10 km² | • It can reduce water level 71 cm | • Increasing floodplain area
• Increasing recreational value | |
| **Wetlands (Mangroves and salt Marshes)** | (Coppenolle, 2018; Gedan et al., 2011) | | | • It can mitigate storm surge 80%
• It can protect against tsunami impacts | • Providing shoreline protection services | |
| **Forest rapirian buffer, basins and ponds and coarse woody debris** | (McVittie et al., 2018) | Pickering, North Yorkshire, UK | 68,6 km² | • Increased water storage 90,000-138,000 m³
• Peak flow rate reduction 6.7–14.7% | • Increase habitat creation value
• Education and knowledge
• Community development
• A benefit/cost ratio of 4.98 | ~€1.58 million |

| Measures | References | Case studies | Area/ volume covered by NBS | Effectiveness | Co-benefits | Cost |
|---|---|---|---|---|---|---|
| | | | | • Flood peak delay by 20 min | | |
| **Renaturation** | (McVittie et al., 2018; NWRM, 2019) | Seymaz river, Switzerland | 0.4 km$^2$ | • Water storage 800,000 m$^3$ | | ~€61 million (€76.3/ m$^3$) |

**iv) Definitions**

**Comments from Referee:** in my opinion, it would be helpful to provide a table with definitions and examples of the various academic terms used in the review: The study provides generic definitions for GI, EbA, and NBS, but it is left upon the reader to interpret the definitions. I would recommend to complement Table 2 with some quantitative figures on water retention, area, costs, advantages, disadvantages etc.... (see also the previous comment).

**Authors' response:** Thank you for the comment and suggestion. Referee 1 also raised similar comments on the definitions. We have included more explanation on the definition in section 3 (now section 2) "Overview of definitions and theoretical backgrounds on the terminology of NBS" Here we also recommend the reader to refer to Nesshöver et al., (2017). and others works for a more exhaustive analysis on terminology, which is beyond the goal of our study. We feel that this additions, together with the newly provided Tables 4 and 5, should provide the reader enough guidance in the interpretation of our results.

**Author's changes in manuscript** (revised and added texts have yellow highlights)**:**

[revised manuscript text omitted]

**v) Drought**

**Comments from Referee** it is well know that land reclamation and restoration reduces evaporation and mitigates the drought risk. However, the authors found only one single study referring to the drought risk. This might be due to a methodology based on "key words" rather than content.

**Authors' response:**

Thank you very much for your suggestion. While the methodology helps limit the scope of the paper, the authors understand that it may also cause some gaps in the study. However, the authors have attempted to review papers as comprehensively as possible to mitigate this issue.

Since the aim of the study is to review Nature Based Solutions and those terms used in conjunction with NBS, additional terminology like reclamation and restoration were not specifically used. This search term could introduce a bias, as the authors then assume the solutions before the review. To be transparent, we have included a sentence to acknowledge that the selection of key search words can limit the hydro-meterological risk measures that appear in returned papers.

Nevertheless, the authors believe that the method still provides useful direction for a state of the art review and defining research gaps on Nature-Based Solutions for Hydro-meteorological risk reduction.

**vi) Scale and examples**

**Comments from Referee** one example that struck me is the NBS "Room for the River Programme" in the Netherlands at the Rhine and Meuse. It is general knowledge that flood protection has to start upstream in the headwaters, where most of the precipitation occurs, to be efficient. Nevertheless, the ms only mentions NBS in the Netherlands (a third of the Netherlands are below sea level and sea levels are rising), ignoring the far more relevant NBS in upstream countries. This might be linked to the somewhat limited methodology of the literature review (see comment i).

**Authors' response**: Thank you very much for your comment. The requirements of this articles that is to focus on peer-reviewed articles in English and we agree with the reviewer that upstream cases exist, which the authors had conducted additional research based on the comment of the reviewer.

It can be summarized that the EU Flood directive specifies that countries upstream or downstream should avoid taking measures that will increase the flood risk to other countries in the same river catchment. In case this is not feasible, the countries should consult with the other member states to agree to the proposed measures (EU, 2019). As far as the authors are aware, there is a project in the Rhine basin called Adaptive Land use for Flood Alleviation

(ALFA). "Room for the River Programme" by the Ducth is also part of this project. However, all of the project's documents from upstream in Germany that the authors have found are only in the grey literatures and in the German language, which are out of the scope of this article.

On the other hand, there are many documents and publications on the "Room for the River Programme" that are available in English. Moreover, "Room for the river programme" is one of the big projects on a large scale NBS which has been successfully studied and implemented it can be used as an example to other countries.

**vii) Tools**

**Comments from Referee:** in my opinion, the review of tools could be shortened, as it is slightly off the topic. Instead, more attention could be given to the quantification of the various benefits of NBS could be provided (see comment iii).

**Authors' response:** Authors have shortened the review of tools. However, the leading message should still be included since the tools are important for selection, evaluation and operation of NBS. One of the purposes of this review is to review the use of techniques, methods and tools for planning, selecting, evaluating and implementing NBS. The benefit of this section is to provide information to the reader as to what the available tools are that can be used for a specific purpose.

The authors have also included some quantification of NBS measures as suggested by the reviewer in Table 4 and 5 as shown in comment iii and benefits of NBS also discuss in section 4.5 in manuscripts. However, discussing the quantitative co-benefits of NBS is still very challenging as there is a lack of information on assessment quantitative value of the ecosystem. Such challenges and limitations will be explictely commented on in the revised version.

**viii) Conclusion**

**Comments from Referee:** the current conclusion provides general and generic statements and any reader somewhat familiar with the topic does not really learn anything new. It would be helpful to generate more conclusive and quantitative statement based on the review: which NBS are most effective, which provide most multi-benefits, which require least areas, which are most accepted?

**Authors' response:** Thank you for the suggestion. The authors have revised the conclusion to summarize the quantitative statement of NBS on; "which NBS are most effective, which provide most multi-benefits, which require the least area, which are the most accepted" as the authors suggested.

However, this has proven to be very difficult because the effectiveness, benefits and acceptance of NBS are dependent on the implementation purposeslocal context and cultural setting. For example, small NBS are more suitable for urban flooding while large scale NBS are more suitable for river floods, coastal floods, droughts and landslides. Large scale NBS can provide more benefits compared to small scale NBS because it has a bigger space, thus more function can be included in the design process. For example, Laojie river project in Taoyuan City in Taiwan changed the channel into an accessible green corridor. This project helps in reducing flood risk, improving riverside landscapes, increasing recreation area, increasing the aesthetic value in the area, and improving river water quality. On the other hand, small scale NBS need less area because most of the measures can be implemented

in the free space. For example, green roofs can be implemented on the roofs of buildings, and permeable pavements can be implemented in car parks. Investments in NBS will benefit society by providing cost-effective measures and adaptive strategies that protect their communities and achieve a range of co-benefits. Therefore, bridging the gaps between researchers, engineers and stakeholders will help to improve the capacity of NBS in reducing hydro-meteorological risk as well as increasing thier benefits. Strengthening these aspects may be beneficial for improving acceptance of NBS at the local level.

In the revised version, all of the above information has been included in the conclusion section, and a summary of quantitative information on effectiveness, co-benefits and cost for different NBS measures can be found in Table 4 and 5 in the revision.

**Track changes document**

[revised manuscript text omitted]

---

## Author Response (AR2)

**Reponses to editor's comments on "Nature-Based Solutions for hydro-meteorological risk reduction: A state-of-the-art review of the research area" by Laddaporn Ruangpan et al.**

I received now the two review reports of your re-vised manuscript "Nature-Based Solutions for hydro-meteorological risk reduction: A state-of-the-art review of the research area".

One reviewer had not further suggestions and the second one addressed several missing papers which should be considered in the review.

**Authors' response:** Thank you for your encouragement and suggestions. Your concerns are addressed in this response letter and the manuscript revised accordingly. Please find our point-by-point response below.

**1. Comments from the editor:** Please consider if it is possible to extend your literature search also for the last months and integrated the newest publication if there are only few publications.

**Authors' response:** The authors have extended literature search for the last months as well as integrated the newest relevant publication into the paper (e.g. page 7 line 1, page 9 line 21 and 25, page 10 line 13, page 15 line 25, page 16 line 7, and page 17 line 4 and line 9)

**2. Comments from the editor:** However, I checked the mentioned special issue (the last mentioned editorial paper) and identified that all papers of the special issue should appear in your search because they were published before December 2018, the journal is included in Scopus and they should meet your search criteria. Therefore, I kindly ask you to clarify this issue.

**Authors' response:** Thank you for the suggestion. The authors have checked the mentioned editorial paper. We see that although the editorial paper and special issues are about Nature-Based Solutions, many articles do not use the keywords ('Nature-Based Solution' or its sister terms) in the title, abstract and keywords, and so do not meet the search criteria. Therefore, these papers were not found in the search result.

Although some articles did meet the search criteria, they were excluded from the review as they focus more on land degradation (e.g., sediment trapping, hydro-geochemical, nutritional, and eco-physiological constraints) than hydro-meteorological risk reduction.

**3. Comments from the editor:** Please also address the comment regarding the Sustainable Development Goals as you addressed this also in your introduction.

**Authors' response:** The authors now have included the Sustainable development Goals (SDG) in Section 4.6 "Multiple benefits of NBS" page 16 line 6.

**Authors' change in the revised manuscript:**

The literature on NBS and its sister concepts increasingly refers to multiple benefits on social, economic and environmental enhancements. The reason for that is that NBS are regarded as sustainable solutions that use ecosystem services to provide multiple benefits for human well-being and the environment, which differs from grey infrastructure. Moreover, these multiple benefits of NBS can help to achieve many of the 2030 Agenda for Sustainable Development Goals (SDGs). The recent publication shows how NBS can contribute to achieving the SDGs (Seifollahi-Aghmiuni et al., 2019). This publication reports that wetland ecosystem services in Sweden positively interact with SDG 1 (no poverty), SDG 2 (zero hunger), SDG 3 (good health and well-being), SDG 6 (clean water and sanitation), SDG 7 (affordable and clean energy), 11 (sustainable cities and communities), SDG 12 (responsible consumption and production), SDG 13 (climate action), SDG 14 (life below water) and SDG 15 (life on land). One of the processes that could provide these benefits is to give more significant consideration to landscape function, adaptive and multi-functionality design (Lennon et al., 2014; Vojinovic et al., 2017), restoring naturally occurring ecosystems and promoting desirable soil (Keesstra et al., 2018).

The literature to date shows that multiple challenges can be continually addressed through NBS. These include reducing flood risk (Song et al., 2018), storing and infiltrating rainfall run-off, delaying and reducing surface runoff, reducing erosion and particulate transport (Loperfido et al., 2014), recharging groundwater discharge, reducing pollution from surface water (Donnell et al., 2018), increasing nutrient retention and removal (Loperfido et al., 2014), maintaining soil moisture, and enhancing vegetation growth. Such benefits help in reaching SDG 6 - ensuring sustainable water management.

Beyond water management, he case for NBS includes their ability to provide additional benefits in improving socio-economic aspects (SDG 11) and human well-being (SDG 3) through recreational areas and aesthetic value (Song et al., 2018), as well as encouraging tourism through the access to nature (Sutton-Grier et al., 2018). Wheeler et al., (2010) quantified the volume and intensity of children's physical activity in greenspace and found that time in greenspace is more likely to lead to greater activity intensity amongst children. The use of NBS can bring economic benefits (SDG 1 and SDG 8) in different ways, such as reduced/prevented damage costs from hydro-meteorological events, energy savings from the reduction of stormwater that typically needs to be treated in a public sewerage system and carbon savings from reduced building energy consumption (heating and cooling) (Soares et al., 2011). Such energy and carbons savings will help contribute to SDG 13.

**4. Comments from the editor:** Furthermore, I highly recommend to address in the discussion the pro and cons of your systematic search design because two reviewers comments on the drawbacks and missing crucial papers.

**Authors' response:** In the revised version, the authors have explained the pros and cons of systematic review in the first paragraph of the conclusion.

**Authors' change in the revised manuscript:**

The present paper provides a critical review of the literature and identifies future research prospects based on the current knowledge gaps in the area of Nature-Based Solutions for hydro-meteorological risk reduction by using a systematic review. The systematic review method helps to limit the scope of the work and also provides useful direction for defining research gaps, as articles can be collected from a board range of sources. However, there are

some disadvantages of systematic reviews. For example, a finite selection of keywords will introduce gaps into the list of articles to be reviewed. Also, important grey literature (e.g. reports and books) could be overlooked. Finally, poorly written abstracts may cause an article to be excluded from the review.

**5. Comments from the editor:** Please find attached the pdf-file with several comments. Please check the format if it is only highlighted in yellow and no further comments are provided

**Authors' response:** In the revised version, authors have cooperated all the comments that suggest by the editor.

**6. Comments from the editor:** I kindly ask you (the second time) to apply the guidelines of NHESS for manuscript preparation (e.g. missing: author contribution, competing interests; physical dimensions and units (e.g. table 4 & 5), figure captions, …).

**Authors' response:** In the revised version, authors have cooperated and carefully applied the guidelines of NHESS for manuscript preparation.

**Authors' change in the revised manuscript:**

*Author contributions.* LR and ZV designed the objectives of the review. LR selected, read and analysed the articles. LR, ZV, SDS and LSL were involved in the production of the paper. LR and ZV have produced the figures. The other authors have contributed to the paper with comments and suggestions. All authors contributed to the writing, editing and revision of the paper.

*Competing interests.* The authors declare that they have no conflict of interest.

**7. Comments from the editor:** In the track-change version I could not find the required changes of language editing as requested for the last version. Please provide here further information

**Authors' response:** The authors have to apologise for this. Changes were made from a native English speaker, and included in the submitted version. However, these changes were mistakenly omitted from the tracked-change version. For clarity, the 'correct' tracked change version of version 3 is included below as well as the tracked change version of version 4.

**The tracked change version of version 4**

**Track changes document of version 4**

[revised manuscript text omitted]

3) Articles published from 1 January 2007 to 1 December 2018.

Sourcing articles from Scopus
(n=1407)

Sourcing articles from Web of Science
(n=1232)

Combining articles from both databases
(n=2639)

Identifying and removing duplicated articles
(n=1395)

**Phase 1**

**Criteria for the selection process**
1) To assess the state-of-the-art in research concerning both small and large scale NBS for hydro-meteorological risk reduction,
2) To review the use of techniques, methods and tools for planning, selecting, evaluating and implementing NBS for hydro-meteorological risk reduction,
3) To review the socio-economic influence in the implementation of NBS for hydro-meteorological risk reduction as well as their co-benefits , effectiveness and costs

Screening articles based on titles and keywords
(n=1204)

Articles excluded after screening
(n=824)
*Reason: no relevant words of search terms*

Evaluating articles based on reading abstracts
(n=380)

Articles excluded after evaluation
(n=195)
*Reason: focus more on other problems than risk*

Articles which had full review
(n=185)

Articles excluded after full review
(n=48)
*Reason: do not meet the objectives of the review*

Articles included in the review
(n=137)

**Phase 2**

Identify knowledge gaps and propose future research prospects

[revised manuscript text omitted]